# Lipschitz-constrained Unsupervised Skill Discovery

**Seohong Park**[1]   **Jongwook Choi**[*2]   **Jaekyeom Kim**[*1]   **Honglak Lee**[2,3]   **Gunhee Kim**[1]

[1]Seoul National University   `{artberryx,jaekyeom,gunhee}@snu.ac.kr`
[2]University of Michigan     `{jwook,honglak}@umich.edu`
[3]LG AI Research

## Abstract

We study the problem of unsupervised skill discovery, whose goal is to learn a set of diverse and useful skills with no external reward. There have been a number of skill discovery methods based on maximizing the mutual information (MI) between skills and states. However, we point out that their MI objectives usually prefer static skills to dynamic ones, which may hinder the application for downstream tasks. To address this issue, we propose *Lipschitz-constrained Skill Discovery* (*LSD*), which encourages the agent to discover more diverse, dynamic, and far-reaching skills. Another benefit of LSD is that its learned representation function can be utilized for solving goal-following downstream tasks even in a zero-shot manner — *i.e.*, without further training or complex planning. Through experiments on various MuJoCo robotic locomotion and manipulation environments, we demonstrate that LSD outperforms previous approaches in terms of skill diversity, state space coverage, and performance on seven downstream tasks including the challenging task of following multiple goals on Humanoid. Our code and videos are available at `https://shpark.me/projects/lsd/`.

## 1 Introduction

Reinforcement learning (RL) aims at learning optimal actions that maximize accumulated reward signals (Sutton & Barto, 2005). Recently, RL with deep neural networks has demonstrated remarkable achievements in a variety of tasks, such as complex robotics control (Gu et al., 2017; Andrychowicz et al., 2020) and games (Schrittwieser et al., 2020; Badia et al., 2020). However, one limitation of the RL framework is that a practitioner has to manually define and tune a reward function for desired behaviors, which is often time-consuming and hardly scalable especially when there are multiple tasks to learn (Hadfield-Menell et al., 2017; Dulac-Arnold et al., 2019).

Therefore, several methods have been proposed to discover *skills* without external task rewards (Gregor et al., 2016; Eysenbach et al., 2019; Sharma et al., 2020), which is often referred to as the *unsupervised skill discovery* problem. Unsupervised discovery of skills helps not only relieve the burden of manually specifying a reward for each behavior, but also provide useful primitives to initialize with or combine hierarchically for solving downstream tasks (Eysenbach et al., 2019; Lee et al., 2020). Moreover, learned skills can effectively demonstrate the agent's capability in the environment, allowing a better understanding of both the agent and the environment.

One of the most common approaches to the unsupervised skill discovery problem is to maximize the mutual information (MI) between skill latent variables and states (Gregor et al., 2016; Achiam et al., 2018; Eysenbach et al., 2019; Hansen et al., 2020; Sharma et al., 2020; Choi et al., 2021; Zhang et al., 2021). Intuitively, these methods encourage a skill latent $z$ to be maximally informative of states or trajectories obtained from a skill policy $\pi(a|s, z)$. As a result, optimizing the MI objective leads to the discovery of diverse and distinguishable behaviors.

However, existing MI-based skill discovery methods share a limitation that they do not necessarily prefer learning 'dynamic' skills (*i.e.*, making large state variations) or task-relevant behaviors such as diverse locomotion primitives. Since MI is invariant to scaling or any invertible transformation of the input variables, there exist infinitely many optima for the MI objective. As such, they will

---

*Equal contribution, listed in alphabetical order.

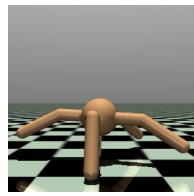 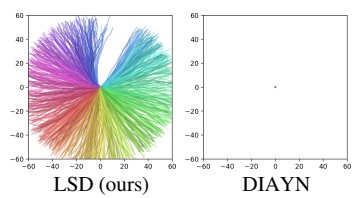 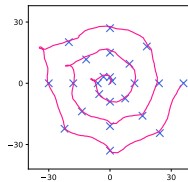

(a) Ant environment. (b) Visualization of discovered 2-D skills on the $x$-$y$ plane. (c) Zero-shot goal following.

Figure 1: Skills discovered by LSD can be used to follow goals with no further training or planning.

converge to the maximum that is most easily optimizable, which would usually be just learning simple and static skills. For instance, Figure 1b and Figure 17 demonstrate that DIAYN (Eysenbach et al., 2019) simply learns to take various postures in place rather than learning locomotion skills in the Ant environment (Schulman et al., 2016). While these works often employ some feature engineering or prior domain knowledge to discover more dynamic skills (*e.g.*, discriminating skills based on $x$-$y$ coordinates only (Eysenbach et al., 2019; Sharma et al., 2020)), it brings about other drawbacks: (i) practitioners need to manually specify the dimensions of interest and (ii) the diversity of skills may be limited to a specific type (*e.g.*, the $x$-$y$ prior results in neglecting non-locomotion behaviors).

In order to address this limitation of MI-based approaches, we propose an unsupervised skill discovery method based on a Lipschitz continuity constraint, named *Lipschitz-constrained Skill Discovery* (*LSD*). Specifically, we argue one reason behind the aforementioned problem is that the MI-based skill discovery methods can easily maximize the MI objective with only slight differences in the state space. To resolve this, we propose a novel objective based on a Lipschitz-constrained state representation function so that the objective maximization in the latent space always entails an increase in traveled distances (or variations) in the state space (Equation (6)).

LSD has several advantages. First, unlike previous MI-based skill discovery objectives, LSD encourages learned skills to have larger traveled distances, which leads to more diverse, dynamic, and far-reaching skills. Second, LSD produces a useful state representation function that can be used to solve goal-following downstream tasks (*i.e.*, reaching multiple goals in order) in a fully *zero-shot* manner (Figure 1c); *i.e.*, with no further training or complex planning. Finally, LSD is easy to implement in contrast to many existing approaches and introduces no additional hyperparameters.

Our contributions can be summarized as follows:

- We propose a novel skill discovery objective based on a Lipschitz constraint named LSD, which maximizes traveled distances in the state space unlike existing MI-based approaches, and thus learns more diverse and dynamic behaviors with no need for feature engineering.

- Since LSD aligns the directions of skills and differences in latent state representations, it can efficiently solve goal-following tasks with a wider range of goals in a zero-shot fashion compared to previous methods, with no burden of additional training or complex planning. Table 1 highlights other distinctive properties of LSD in comparison to existing approaches.

- LSD exhibits the best performance in terms of the state space coverage on five MuJoCo environments and final rewards on seven downstream tasks, including AntMultiGoals (Kim et al., 2021), HumanoidMultiGoals and FetchPushGoal, compared to previous skill discovery methods such as DIAYN (Eysenbach et al., 2019), DADS (Sharma et al., 2020) and IBOL (Kim et al., 2021).

## 2 PRELIMINARIES AND RELATED WORK

### 2.1 PROBLEM SETTING

We consider a Markov decision process (MDP) $\mathcal{M} = (\mathcal{S}, \mathcal{A}, p)$ without external rewards, where $\mathcal{S}$ is a (continuous) state space, $\mathcal{A}$ is an action space, and $p(s_{t+1}|s_t, a_t)$ is a stochastic transition dynamics function. We represent a *skill* with a *latent variable* $z \in \mathcal{Z}$ and a latent-conditioned policy $\pi(a|s, z)$. The skill latent space $\mathcal{Z}$ can be either discrete or continuous; we use $N$ to denote the number of skills in the discrete case, and $d$ to denote the dimensionality of the skill latent space in the continuous case. Given a skill $z$ and a skill policy $\pi(a|s, z)$, a trajectory $\tau = (s_0, a_0, \ldots, s_T)$ is sampled with the following generative process: $p^\pi(\tau|z) = p(s_0) \prod_{t=0}^{T-1} \pi(a_t|s_t, z) p(s_{t+1}|s_t, a_t)$.

Table 1: Comparison of unsupervised skill discovery methods. Refer to Section 2.2 for citations.

| Property | VIC | DIAYN | DADS | VISR | EDL | APS | IBOL | LSD (ours) |
|---|---|---|---|---|---|---|---|---|
| Prefer 'dynamic' skills | ✗ | ✗ | ✗ | ✗ | ✓ | ✓ | ✓ | ✓ |
| Provide dense reward | ✗ | ✓ | ✓ | ✓ | ✓ | ✓ | ✗ | ✓ |
| Discover continuous skills | ✓ | ✓ | ✓ | ✓ | ✓ | ✓ | ✓ | ✓ |
| Discover discrete skills | ✓ | ✓ | ✓ | ✗ | ✓ | ✗ | ✗ | ✓ |
| Zero-shot goal-following | ✓[*] | ✓[*] | ✓[†] | ✓[*] | ✓[*] | ✓[*] | ✗ | ✓ |

Properties — *"Prefer 'dynamic' skills"*: whether the algorithm prefers skills beyond simple and static ones. *"Zero-shot goal-following"*: whether learned skills can be used for following multiple goals (from an arbitrary state) without additional training, where † denotes that it still needs planning and * denotes that its skill discriminator may not cope well with unseen goals or initial states.

We use uppercase letters to denote random variables, and $h(\cdot)$ and $I(\cdot; \cdot)$ to represent the differential entropy and the mutual information, respectively.

## 2.2 PRIOR WORK ON UNSUPERVISED SKILL DISCOVERY

A number of previous methods maximize $I(Z; S)$ to learn diverse skills. One line of research employs the identity $I(Z; S) = h(Z) - h(Z|S) \geq \mathbb{E}_{z \sim p(z), s \sim p^\pi(s|z)}[\log q(z|s)] - \mathbb{E}_{z \sim p(z)}[\log p(z)]$ or its variants, where the skill discriminator $q(z|s)$ is a variational approximation of the posterior $p(z|s)$ (Barber & Agakov, 2003). VIC (Gregor et al., 2016) maximizes the MI between the last states and skills given the initial state. DIAYN (Eysenbach et al., 2019) optimizes the MI between individual states and skills. VALOR (Achiam et al., 2018) also takes a similar approach, but considers the whole trajectories instead of states. VISR (Hansen et al., 2020) models the variational posterior as the von-Mises Fisher distribution, which results in an inner-product reward form and hence enables combining with successor features (Barreto et al., 2017). HIDIO (Zhang et al., 2021) examines multiple variants of the MI identity, and jointly learns skills with a hierarchical controller that maximizes the task reward. Choi et al. (2021) point out the equivalency between the MI-based objective and goal-conditioned RL, and show that Spectral Normalization (Miyato et al., 2018) improves the quality of learned skills. DADS (Sharma et al., 2020) maximizes the opposite direction of the mutual information identity $I(Z; S) = h(S) - h(S|Z)$ with the skill dynamics model $q(s_{t+1}|s_t, z)$, which allows zero-shot planning on downstream tasks. However, these methods share a limitation: they do not always prefer to reach distant states or to learn dynamic skills, as we can maximize $I(Z; S)$ even with the smallest state variations. One possible way to address this issue is to use heuristics such as feature engineering; for example, the $x$-$y$ prior (Sharma et al., 2020) enforces skills to be discriminated only by their $x$-$y$ coordinates so that the agent can discover locomotion skills.

On the other hand, a couple of methods overcome this limitation by integrating with exploration techniques. EDL (Campos Camúñez et al., 2020) first maximizes the state entropy $h(S)$ with SMM exploration (Lee et al., 2019), and then encodes the discovered states into skills via VAE (Kingma & Welling, 2014). APS (Liu & Abbeel, 2021a) combines VISR (Hansen et al., 2020) with APT (Liu & Abbeel, 2021b), an exploration method based on $k$-nearest neighbors. Yet, we empirically confirm that such a pure exploration signal is insufficient to make large and consistent transitions in states. IBOL (Kim et al., 2021) takes a hierarchical approach where it first pre-trains a low-level policy to make reaching remote states easier, and subsequently learns a high-level skill policy based on the information bottleneck framework (Tishby et al., 2000). While IBOL can discover skills reaching distant states in continuous control environments without locomotion priors (*e.g.*, $x$-$y$ prior), it still has some limitations in that (1) IBOL cannot discover discrete skills, (2) it still capitalizes on input feature engineering in that they exclude the locomotion coordinates from the low-level policy, and (3) it consists of a two-level hierarchy with several additional hyperparameters, which make the implementation difficult. On the contrary, our proposed LSD can discover diverse skills in both discrete and continuous settings without using any feature engineering, and is easy to implement as it requires no additional hyperparameters or hierarchy. Table 1 overviews the comparison of properties between different skill discovery methods.

## 3 LIPSCHITZ-CONSTRAINED SKILL DISCOVERY (LSD)

We first analyze limitations of existing MI-based methods for unsupervised skill discovery (Section 3.1), and then derive our approach for learning continuous skills, *Lipschitz-constrained Skill*

*Discovery* (*LSD*), which encourages the agent to have large traveled distances in the state space (Section 3.2). We also show how learned skills can be used to solve goal-following tasks in a zero-shot fashion (Section 3.3), and extend LSD to discovery of discrete skills (Section 3.4).

## 3.1 LIMITATION OF MI-BASED METHODS

Before deriving our objective for discovery of continuous skills, we review variational MI-based skill discovery algorithms and discuss why such methods might end up learning only simple and static skills. The MI objective $I(Z; S)$ with continuous skills can be written with the variational lower bound as follows (Eysenbach et al., 2019; Choi et al., 2021):

$$I(Z; S) = -h(Z|S) + h(Z) = \mathbb{E}_{z \sim p(z), s \sim p^\pi(s|z)}[\log p(z|s) - \log p(z)] \tag{1}$$

$$\geq \mathbb{E}_{z,s}[\log q(z|s)] + (\text{const}) = -\frac{1}{2} \mathbb{E}_{z,s} \left[ \|z - \mu(s)\|^2 \right] + (\text{const}), \tag{2}$$

where we assume that a skill $z \in \mathbb{R}^d$ is sampled from a fixed prior distribution $p(z)$, and $q(z|s)$ is a variational approximation of $p(z|s)$ (Barber & Agakov, 2003; Mohamed & Rezende, 2015), parameterized as a normal distribution with unit variance, $\mathcal{N}(\mu(s), \mathbf{I})$ (Choi et al., 2021). Some other methods are based on a conditional form of mutual information (Gregor et al., 2016; Sharma et al., 2020); for instance, the objective of VIC (Gregor et al., 2016) can be written as

$$I(Z; S_T|S_0) \geq \mathbb{E}_{z,\tau}[\log q(z|s_0, s_T)] + (\text{const}) = -\frac{1}{2} \mathbb{E}_{z,\tau} \left[ \|z - \mu(s_0, s_T)\|^2 \right] + (\text{const}), \tag{3}$$

where we assume that $p(z|s_0) = p(z)$ is a fixed prior distribution, and the posterior is chosen as $q(z|s_0, s_T) = \mathcal{N}(\mu(s_0, s_T), \mathbf{I})$ in a continuous skill setting.

One issue with Equations (2) and (3) is that these objectives can be fully maximized even with small differences in states as long as different $z$'s correspond to even marginally different $s_T$'s, not necessarily encouraging more 'interesting' skills. This is especially problematic because discovering skills with such slight or less dynamic state variations is usually a 'lower-hanging fruit' than making dynamic and large differences in the state space (*e.g.*, $\mu$ can simply map the angles of Ant's joints to $z$). As a result, continuous DIAYN and DADS discover only posing skills on Ant (Figures 2a and 17) in the absence of any feature engineering or tricks to elicit more diverse behaviors. We refer to Appendix H for quantitative demonstrations of this phenomenon on MuJoCo environments.

**Further decomposition of the MI objective.** Before we address this limitation, we decompose the objective of VIC (Equation (3)) to get further insights that will inspire our new objective in Section 3.2. Here, we model $\mu(s_0, s_T)$ with $\phi(s_T) - \phi(s_0)$ to focus on the relative differences between the initial and final states, where $\phi : \mathcal{S} \to \mathcal{Z}$ is a learnable *state representation function* that maps a state observation into a latent space, which will be the core component of our method. This choice makes the latent skill $z$ represent a direction or displacement in the latent space induced by $\phi$. Then, we can rewrite Equation (3) as follows:

$$\mathbb{E}_{z,\tau} \left[ \log q(z|s_0, s_T) \right] + (\text{const}) = -\frac{1}{2} \mathbb{E}_{z,\tau} \left[ \|z - (\phi(s_T) - \phi(s_0))\|^2 \right] + (\text{const}) \tag{4}$$

$$= \underbrace{\mathbb{E}_{z,\tau} \left[ (\phi(s_T) - \phi(s_0))^\top z \right]}_{\text{① alignment of the direction}} - \frac{1}{2} \underbrace{\mathbb{E}_{z,\tau} \left[ \|\phi(s_T) - \phi(s_0)\|^2 \right]}_{\text{② implicit regularizer}} + (\text{const}), \tag{5}$$

where we use the fact that $\mathbb{E}_z[z^\top z]$ is a constant as $p(z)$ is a fixed distribution. This decomposition of the MI lower-bound objective provides an intuitive interpretation: the first inner-product term ① in Equation (5) encourages the direction vector $\phi(s_T) - \phi(s_0)$ to be aligned with $z$, while the second term ② regularizes the norm of the vector $\phi(s_T) - \phi(s_0)$.

## 3.2 CONTINUOUS LSD

**The LSD objective.** We now propose our new objective that is based on neither a skill discriminator (Equation (3)) nor mutual information but a Lipschitz-constrained state representation function, in order to address the limitation that $I(Z; S)$ can be fully optimized with small state differences. Specifically, inspired by the decomposition in Equation (5), we suggest using the directional term ① as our objective. However, since this term alone could be trivially optimized by just increasing the value of $\phi(s_T)$ to the infinity regardless of the final state $s_T$, we apply the 1-Lipschitz constraint on the state representation function $\phi$ so that maximizing our objective in the latent space can result

in an increase in state differences. This leads to a constrained maximization objective as follows:

$$J^{\text{LSD}} = \mathbb{E}_{z,\tau} \left[ (\phi(s_T) - \phi(s_0))^\top z \right] \quad \text{s.t.} \quad \forall x, y \in \mathcal{S} \quad \|\phi(x) - \phi(y)\| \leq \|x - y\|, \quad (6)$$

where $J^{\text{LSD}}$ is the objective of our proposed ***Lipschitz-constrained Skill Discovery*** (**LSD**).

The LSD objective encourages the agent to prefer skills with larger traveled distances, unlike previous MI-based methods, as follows. First, in order to maximize the inner product in Equation (6), the length of $\phi(s_T) - \phi(s_0)$ should be increased. It then makes its upper bound $\|s_T - s_0\|$ increase as well due to the 1-Lipschitz constraint (*i.e.*, $\|\phi(s_T) - \phi(s_0)\| \leq \|s_T - s_0\|$). As a result, it leads to learning more dynamic skills in terms of state differences.

Note that LSD's objective differs from VIC's in an important way. Equation (3) tries to equate the value of $z$ and $\mu(s_0, s_T)$ (*i.e.*, it tries to recover $z$ from its skill discriminator), while the objective of LSD (Equation (6)) only requires the directions of $z$ and $\phi(s_T) - \phi(s_0)$ to be aligned.

We also note that our purpose of enforcing the Lipschitz constraint is very different from its common usages in machine learning. Many works have adopted the Lipschitz continuity to regularize functions for better generalization (Neyshabur et al., 2018; Sokolic et al., 2017), interpretability (Tsipras et al., 2018) or stability (Choi et al., 2021). On the other hand, we employ it to ensure that maximization of the reward entails increased state variations. The Lipschitz constant 1 is chosen empirically as it can be easily implemented using Spectral Normalization (Miyato et al., 2018).

**Per-step transition reward.** By eliminating the second term in Equation (5), we can further decompose the objective using a telescoping sum as

$$J^{\text{LSD}} = \mathbb{E}_{z,\tau} \left[ (\phi(s_T) - \phi(s_0))^\top z \right] = \mathbb{E}_{z,\tau} \left[ \sum_{t=0}^{T-1} (\phi(s_{t+1}) - \phi(s_t))^\top z \right]. \quad (7)$$

This formulation enables the optimization of $J^{\text{LSD}}$ with respect to the policy $\pi$ (*i.e.*, reinforcement learning steps) with a *per-step* transition reward given as:

$$r_z^{\text{LSD}}(s_t, a_t, s_{t+1}) = (\phi(s_{t+1}) - \phi(s_t))^\top z. \quad (8)$$

Compared to the per-trajectory reward in Equation (3) (Gregor et al., 2016), this can be optimized more easily and stably with off-the-shelf RL algorithms such as SAC (Haarnoja et al., 2018a).

**Connections to previous methods.** The per-step reward function of Equation (8) is closely related to continuous DIAYN (Eysenbach et al., 2019; Choi et al., 2021) and VISR (Hansen et al., 2020):

$$r_z^{\text{DIAYN}}(s_t, a_t, s_{t+1}) = \log q^{\text{DIAYN}}(z|s_{t+1}) \propto -\|\phi(s_{t+1}) - z\|^2 + (\text{const}) \quad (9)$$

$$r_{\tilde{z}}^{\text{VISR}}(s_t, a_t, s_{t+1}) = \log q^{\text{VISR}}(\tilde{z}|s_t) \propto \tilde{\phi}(s_t)^\top \tilde{z} + (\text{const}) \quad (10)$$

$$r_z^{\text{LSD}}(s_t, a_t, s_{t+1}) = (\phi(s_{t+1}) - \phi(s_t))^\top z, \quad (11)$$

where $\tilde{z}$ and $\tilde{\phi}(s_t)$ in VISR are the normalized vectors of unit length (*i.e.*, $z/\|z\|$ and $\phi(s_t)/\|\phi(s_t)\|$), respectively. We assume that $q^{\text{DIAYN}}$ is parameterized as a normal distribution with unit variance, and $q^{\text{VISR}}$ as a von-Mises Fisher (vMF) distribution with a scale parameter of 1 (Hansen et al., 2020).

While it appears that there are some similarities among Equation (9)–(11), LSD's reward function is fundamentally different from the others in that it optimizes neither a log-probability nor mutual information, and thus only LSD seeks for distant states. For instance, VISR (Equation (10)) has the most similar form as it also uses an inner product, but $\phi(s_{t+1}) - \phi(s_t)$ in $r_z^{\text{LSD}}$ does not need to be a unit vector unlike VISR optimizing the vMF distribution. Instead, LSD increases the difference of $\phi$ to encourage the agent to reach more distant states. In addition, while Choi et al. (2021) also use Spectral Normalization, our objective differs from theirs as we do not optimize $I(Z; S)$. We present further discussion and an empirical comparison of reward functions in Appendix D.

**Implementation.** In order to maximize the LSD objective (Equation (8)), we alternately train $\pi$ with SAC (Haarnoja et al., 2018a) and $\phi$ with stochastic gradient descent. We provide the full procedure for LSD in Appendix C. Note that LSD has the same components and hyperparameters as DIAYN, and is thus easy to implement, especially as opposed to IBOL (Kim et al., 2021), a state-of-the-art skill discovery method on MuJoCo environments that requires a two-level hierarchy with many moving components and hyperparameters.

### 3.3 ZERO-SHOT SKILL SELECTION

Another advantage of the LSD objective $J^{\text{LSD}}$ (Equation (6)) is that the learned state representation $\phi(s)$ allows solving goal-following downstream tasks in a *zero-shot* manner (*i.e.*, without any further training or complex planning), as $z$ is aligned with the direction in the representation space. Although it is also possible for DIAYN-like methods to reach a single goal from the initial state in a zero-shot manner (Choi et al., 2021), LSD is able to reach a goal from an arbitrary state or follow multiple goals thanks to the directional alignment without any additional modifications to the method. Specifically, if we want to make a transition from the current state $s \in \mathcal{S}$ to a target state $g \in \mathcal{S}$, we can simply repeat selecting the following $z$ until reaching the goal:

$$z = \alpha(\phi(g) - \phi(s)) \, / \, \|\phi(g) - \phi(s)\|, \tag{12}$$

and executing the latent-conditioned policy $\pi(a|s,z)$ to choose an action. Here, $\alpha$ is a hyperparameter that controls the norm of $z$, and we find that $\alpha \approx \mathbb{E}_{z \sim p(z)}[\|z\|]$ empirically works the best (*e.g.*, $\alpha = 2^{-\frac{1}{2}}\Gamma(1/2) \approx 1.25$ for $d = 2$). As will be shown in Section 4.3, this zero-shot scheme provides a convenient way to immediately achieve strong performance on many goal-following downstream tasks. Note that, in such tasks, this scheme is more efficient than the zero-shot *planning* of DADS with its learned skill dynamics model (Sharma et al., 2020); LSD does not need to learn models or require any planning steps in the representation space.

### 3.4 DISCRETE LSD

Continuous LSD can be extended to discovery of *discrete* skills. One might be tempted to use the one-hot encoding for $z$ with the same objective as continuous LSD (Equation (6)), as in prior methods (Eysenbach et al., 2019; Sharma et al., 2020). For discrete LSD, however, we cannot simply use the standard one-hot encoding. This is because an encoding that has a *non-zero* mean could make all the skills *collapse* into a single behavior in LSD. For example, without loss of generality, suppose the first dimension of the mean of the $N$ encoding vectors is $c > 0$. Then, if the agent finds a final state $s_T$ that makes $\|s_T - s_0\|$ fairly large, it can simply learn the skill policy to reach $s_T$ regardless of $z$ so that the agent can always receive a reward of $c \cdot \|s_T - s_0\|$ on average, by setting, *e.g.*, $\phi(s_T) = [\|s_T - s_0\|, 0, \ldots, 0]^\top$ and $\phi(s_0) = [0, 0, \ldots, 0]$. In other words, the agent can easily exploit the reward function without learning any diverse set of skills.

To resolve this issue, we propose using a zero-centered one-hot vectors as follows:

$$Z \sim \text{unif}\{z_1, z_2, \ldots, z_N\} \quad \text{where } [z_i]_j = \begin{cases} 1 & \text{if } i = j \\ -\frac{1}{N-1} & \text{otherwise} \end{cases} \text{ for } i, j \in \{1, \ldots, N\}, \tag{13}$$

where $N$ is the number of skills, $[\cdot]_i$ denotes the $i$-th element of the vector, and $\text{unif}\{\ldots\}$ denotes the uniform distribution over a set. Plugging into Equation (8), the reward for the $k$-th skill becomes

$$r_k^{\text{LSD}}(s_t, a_t, s_{t+1}) = [\phi(s_{t+1}) - \phi(s_t)]_k - \frac{1}{N-1} \sum_{i \in \{1,2,\ldots,N\}\setminus\{k\}} [\phi(s_{t+1}) - \phi(s_t)]_i. \tag{14}$$

This formulation provides an intuitive interpretation: it enforces the $k$-th element of $\phi(s)$ to be the only indicator for the $k$-th skill in a *contrastive* manner. We note that Equation (14) consistently pushes the difference in $\phi$ to be as large as possible, unlike prior approaches using the softmax function (Eysenbach et al., 2019; Sharma et al., 2020). Thanks to the Lipschitz constraint on $\phi(s)$, this makes the agent learn diverse and dynamic behaviors as in the continuous case.

## 4 EXPERIMENTS

We compare LSD with multiple previous skill discovery methods on various MuJoCo robotic locomotion and manipulation environments (Todorov et al., 2012; Schulman et al., 2016; Plappert et al., 2018) from OpenAI Gym (Brockman et al., 2016). We aim to answer the following questions: (i) How well does LSD discover skills on high-dimensional continuous control problems, compared to previous approaches? (ii) Can the discovered skills be used for solving goal-following tasks in a zero-shot fashion? In Appendix, we present an ablation study for demonstrating the importance of LSD's components (Appendix D) and analyses of learned skills (Appendix E).

**Experimental setup.** We make evaluations on three MuJoCo robotic locomotion environments (Ant, Humanoid, HalfCheetah) and three robotic manipulation environments (FetchPush, FetchSlide and FetchPickAndPlace). Following the practice in previous works (Sharma et al., 2020; Kim et al.,

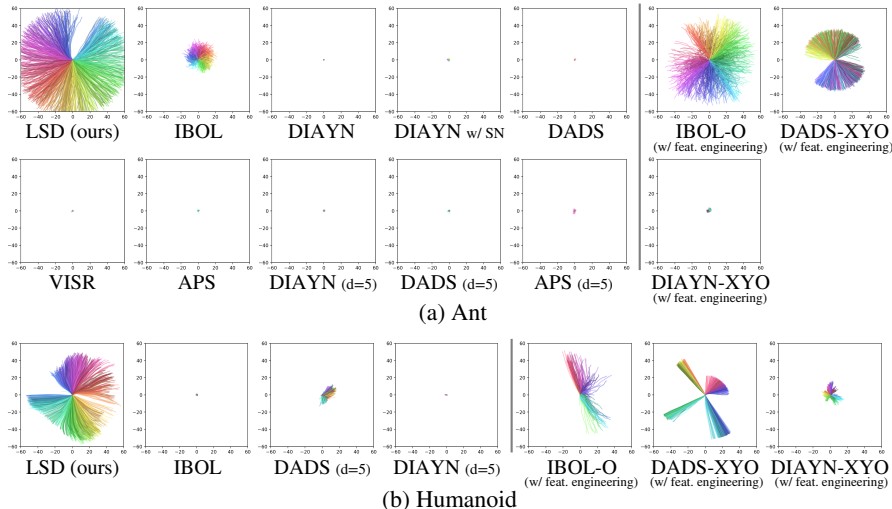

Figure 2: Visualization of 2-D (or 5-D) continuous skills discovered by various methods. We plot the $x$-$y$ trajectories of the agent. Each color represents the direction of the skill latent variable $z$. See Figure 13 for a zoomed-in version.

2021), we mainly compare skill discovery methods on Ant, but we additionally adopt Humanoid for quantitative comparisons with competitive baselines, since it is often considered the most challenging environment in the MuJoCo benchmark. On the manipulation environments, we compare LSD to baselines combined with MUSIC-u (Zhao et al., 2021), an intrinsic reward that facilitates the agent to have control on target objects. For continuous skills, unless otherwise mentioned, we use two-dimensional skill latents ($d = 2$) sampled from the standard normal distribution, following Kim et al. (2021). On the locomotion environments, we normalize the state dimensions to ensure that the different scales of the dimensions have less effect on skill discovery. We repeat all the experiments eight times and denote their 95% confidence intervals with shaded areas or error bars. We refer to Appendix I for the full experimental details.

**Baseline methods.** We make comparisons with six skill discovery algorithms: DIAYN (Eysenbach et al., 2019), DADS (Sharma et al., 2020), VISR (Hansen et al., 2020), EDL (Campos Camúñez et al., 2020), APS (Liu & Abbeel, 2021a) and IBOL (Kim et al., 2021).

Additionally, we consider two variants of skill discovery methods: the $x$-$y$ prior (denoted with the suffix '-XY') and $x$-$y$ omission (with '-O') (Sharma et al., 2020; Kim et al., 2021). The $x$-$y$ prior variant restricts the input to the skill discriminator or the dynamics model only to the positional information, enforcing the agent to discover locomotion skills. The $x$-$y$ omission variant excludes the locomotion coordinates from the input to policies or dynamics models (but not to discriminators) to impose an inductive bias that the agent can choose actions regardless of its location. We denote the variants that have both modifications with the suffix '-XYO'. While previous methods mostly require such feature engineering or tricks to discover skills that move consistently or have large variations in the state space, we will demonstrate that LSD can discover diverse skills on MuJoCo locomotion environments without using hand-engineered features.

## 4.1 SKILLS LEARNED WITH CONTINUOUS LSD

**Visualization of skills.** We train LSD and baselines on the Ant environment to learn two-dimensional continuous skills ($d = 2$). Figure 2a visualizes the learned skills as trajectories of the Ant agent on the $x$-$y$ plane. LSD discovers skills that move far from the initial location in almost all possible directions, while the other methods except IBOL fail to discover such locomotion primitives without feature engineering (*i.e.*, $x$-$y$ prior) even with an increased skill dimensionality ($d = 5$). Instead, they simply learn to take static postures rather than to move; such a phenomenon is also reported in Gu et al. (2021). This is because their MI objectives do not particularly induce the agent to increase state variations. On the other hand, LSD discovers skills reaching even farther than those of the baselines using feature engineering (IBOL-O, DADS-XYO and DIAYN-XYO).

Figure 2b demonstrates the results on the Humanoid environment. As in Ant, LSD learns diverse skills walking or running consistently in various directions, while skills discovered by other methods

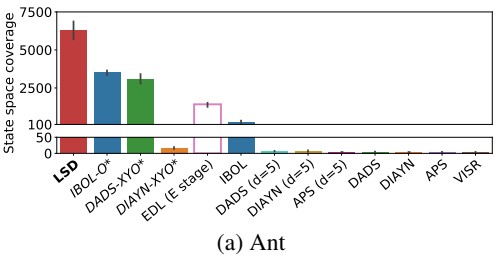
(a) Ant

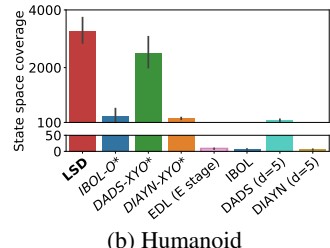
(b) Humanoid

Figure 3: Plots of state space coverage. Asterisks (*) denote the methods with feature engineering.

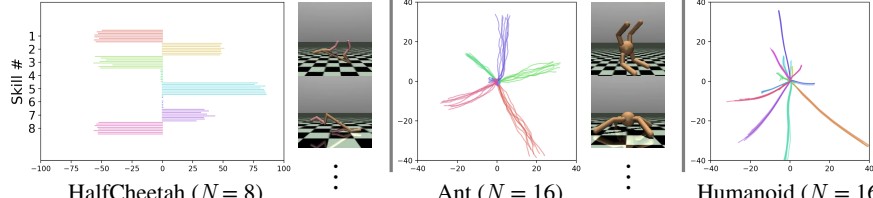

HalfCheetah ($N = 8$)     Ant ($N = 16$)     Humanoid ($N = 16$)

Figure 4: Qualitative results of discrete LSD (Section 4.2). We visualize each skill's trajectories on the $x$ axis (HalfCheetah) or the $x$-$y$ plane (Ant and Humanoid). See Appendix J for more results.

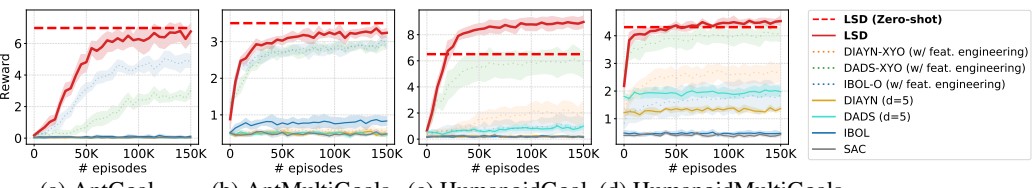

(a) AntGoal    (b) AntMultiGoals    (c) HumanoidGoal    (d) HumanoidMultiGoals

Figure 5: Performance on downstream tasks after skill discovery (Section 4.3).

are limited in terms of the state space coverage or the variety of directions. We provide the videos of skills discovered by LSD at https://shpark.me/projects/lsd/.

**Quantitative evaluation.** For numerical comparison, we measure the *state space coverage* (Kim et al., 2021) of each skill discovery method on Ant and Humanoid. The state space coverage is measured by the number of occupied $1 \times 1$ bins on the $x$-$y$ plane from 200 randomly sampled trajectories, averaged over eight runs. For EDL, we report the state space coverage of its SMM exploration phase (Lee et al., 2019). Figure 3 shows that on both environments, LSD outperforms all the baselines even including those with feature engineering.

## 4.2 SKILLS LEARNED WITH DISCRETE LSD

We train discrete LSD on Ant, HalfCheetah and Humanoid with $N = 6$, 8, 16, where $N$ is the number of skills. While continuous LSD mainly discovers locomotion skills, we observe that discrete LSD learns more diverse skills thanks to its contrastive scheme (Figure 4, Appendix J). On Ant, discrete LSD discovers a skill set consisting of five locomotion skills, six rotation skills, three posing skills and two flipping skills. On HalfCheetah, the agent learns to run forward and backward in multiple postures, to roll forward and backward, and to take different poses. Finally, Humanoid learns to run or move in multiple directions and speeds with unique gaits. We highly recommend the reader to check the videos available on our project page. We refer to Appendix G.1 for quantitative evaluations. To the best of our knowledge, LSD is the only method that can discover such diverse and dynamic behaviors (*i.e.*, having large traveled distances in many of the state dimensions) within a single set of skills in the absence of feature engineering.

## 4.3 COMPARISON ON DOWNSTREAM TASKS

As done in Eysenbach et al. (2019); Sharma et al. (2020); Kim et al. (2021), we make comparisons on downstream goal-following tasks to assess how well skills learned by LSD can be employed for solving tasks in a hierarchical manner, where we evaluate our approach not only on AntGoal and AntMultiGoals (Eysenbach et al., 2019; Sharma et al., 2020; Kim et al., 2021) but also on the more challenging tasks: HumanoidGoal and HumanoidMultiGoals. In the '-Goal' tasks, the agent should

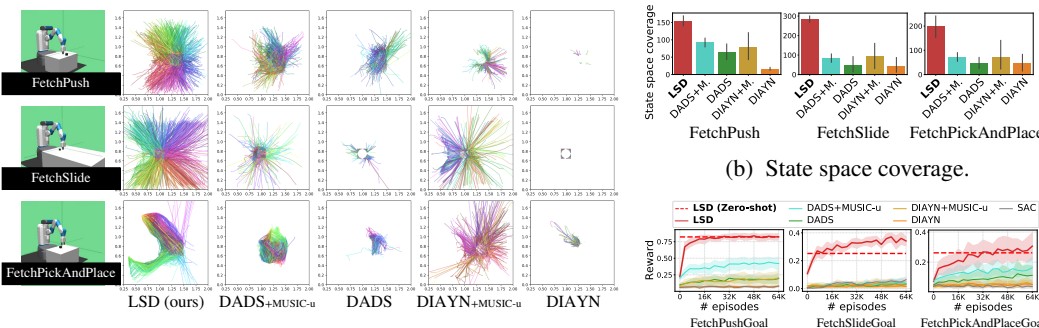

(a) Visualization of trajectories of the target object on the $x$-$y$ plane (*i.e.*, the table) in three Fetch manipulation environments. Each color represents the direction of the skill latent variable $z$.

(b) State space coverage.

(c) Average reward on downstream tasks.

Figure 6: Comparisons on the Fetch robotic manipulation environments (Section 4.4).

reach a uniformly sampled random goal on the $x$-$y$ plane, while in the '-MultiGoals' tasks, the agent should navigate through multiple randomly sampled goals in order. The agent is rewarded only when it reaches a goal. We refer to Appendix I for the full details of the tasks.

We first train each skill discovery method with continuous skills (without rewards), and then train a hierarchical meta-controller on top of the learned skill policy (kept frozen) with the task reward. The meta-controller observes the target goal concatenated to the state observation. At every $K$-th step, the controller selects a skill $z \in [-2, 2]^d$ to be performed for the next $K$ steps, and the chosen skill is executed by the skill policy for the $K$ steps. We also examine zero-shot skill selection of LSD, denoted 'LSD (Zero-shot)', where the agent chooses $z$ at every step according to Equation (12).

**Results.** Figure 5 shows the performance of each algorithm evaluated on the four downstream tasks. LSD demonstrates the highest reward in all of the environments, outperforming even the baselines with feature engineering. On top of that, in some environments such as AntMultiGoals, LSD's zero-shot skill selection performs the best, while still exhibiting strong performance on the other tasks. From these results, we show that LSD is capable of solving downstream goal-following tasks very efficiently with no further training or complex planning procedures.

## 4.4 EXPERIMENTS ON ROBOTIC MANIPULATION ENVIRONMENTS

In order to demonstrate that LSD can also discover useful skills in environments other than locomotion tasks, we make another evaluation on three Fetch robotic manipulation environments (Plappert et al., 2018). We compare LSD with other skill discovery methods combined with MUSIC-u (Zhao et al., 2021), an intrinsic reward that maximizes the mutual information $I(S^a; S^s)$ between the agent state $S^a$ and the surrounding state $S^s$. For a fair comparison with MUSIC, we make use of the same prior knowledge for skill discovery methods including LSD to make them focus only on the surrounding states (Zhao et al., 2021), which correspond to the target object in our experiments.

**Results.** Figure 6a visualizes the target object's trajectories of 2-D continuous skills learned by each algorithm. They suggest that LSD can control the target object in the most diverse directions. Notably, in FetchPickAndPlace, LSD learns to pick the target object in multiple directions without any task reward or intrinsic motivation like MUSIC-u. Figures 6b and 6c present quantitative evaluations on the Fetch environments. As in Ant and Humanoid, LSD exhibits the best state space coverage (measured with $0.1 \times 0.1$-sized bins) in all the environments. Also, LSD and LSD (Zero-shot) outperform the baselines by large margins on the three downstream tasks.

## 5 CONCLUSION

We presented LSD as an unsupervised skill discovery method based on a Lipschitz constraint. We first pointed out the limitation of previous MI-based skill discovery objectives that they are likely to prefer static skills with limited state variations, and resolved this by proposing a new objective based on a Lipschitz constraint. It resulted in learning more dynamic skills and a state representation function that enables zero-shot skill selection. Through multiple quantitative and qualitative experiments on robotic manipulation and complex locomotion environments, we showed that LSD outperforms previous skill discovery methods in terms of the diversity of skills, state space coverage and performance on downstream tasks. Finally, we refer the readers to Appendix A for a discussion of limitations and future directions.

## ACKNOWLEDGEMENTS

We thank the anonymous reviewers for their helpful comments. This work was supported by Samsung Advanced Institute of Technology, Brain Research Program by National Research Foundation of Korea (NRF) (2017M3C7A1047860), Institute of Information & communications Technology Planning & Evaluation (IITP) grant funded by the Korea government (MSIT) (No. 2019-0-01082, SW StarLab), Institute of Information & communications Technology Planning & Evaluation (IITP) grant funded by the Korea government (MSIT) (No. 2021-0-01343, Artificial Intelligence Graduate School Program (Seoul National University)) and NSF CAREER IIS-1453651. J.C. was partly supported by Korea Foundation for Advanced Studies. J.K. was supported by Google PhD Fellowship. Gunhee Kim is the corresponding author.

## REPRODUCIBILITY STATEMENT

We make our implementation publicly available in the repository at https://vision.snu.ac.kr/projects/lsd/ and provide the full implementation details in Appendix I.

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

APPENDIX: LIPSCHITZ-CONSTRAINED UNSUPERVISED SKILL DISCOVERY

## A    LIMITATIONS AND FUTURE DIRECTIONS

LSD may not encourage dynamic behaviors in some environments where Lipschitz constraints are not semantically meaningful, such as control from pixel observations. This issue is addressable by incorporating representation learning, which we leave for future work. Also, in contrast to discrete LSD that learns more diverse skills, continuous LSD mainly discovers locomotion skills that move as far as possible, since its objective is independent of the magnitude of $z$. While this has an advantage in that we can later choose skills just by their directions (enabling zero-shot skill selection), making LSD respect the magnitude as well can be another interesting research direction. Finally, while LSD does not use any explicit feature engineering, we note that the skills LSD learns are still affected by the shape of maximally reachable regions in the (normalized) state space.

## B    EXTENDED RELATED WORK

The LSD reward (Equation (8)) might look very similar to the objective of *eigenoptions* (Machado et al., 2017; 2018): $(\phi(s_{t+1}) - \phi(s_t))^\top e$, where $\phi$ is a *fixed* (or pre-trained) representation function of states and $e$ is an eigenvector of the successor representation matrix (Dayan, 1993) computed from a *fixed random* policy. They define eigenoptions as the options (or skills) that maximize this reward for each of the eigenvectors $e$ of the $N$ largest eigenvalues. However, our interest differs from their setting, since we *learn* both the policy and the representation function in order to seek diverse and dynamic skills, and our approach is applicable to continuous skill settings as well as discrete skill learning.

## C    TRAINING PROCEDURE FOR LSD

---

**Algorithm 1:** Lipschitz-constrained Skill Discovery (LSD)

---

Initialize skill policy $\pi$ and representation function $\phi$;
**while** *not converged* **do**
    **for** $i = 1, \ldots, $ *(# episodes per epoch)* **do**
        Sample skill $z$ from $p(z)$;
        Sample trajectory (episode) $\tau$ with $\pi(\cdot|\cdot, z)$ and $z$;
    Compute reward $r^{\text{LSD}}(s_t, a_t, s_{t+1}) = (\phi(s_{t+1}) - \phi(s_t))^\top z$  (Equation (8));
    Update $\phi$ using SGD to maximize Equation (8) under Spectral Normalization;
    Update $\pi$ using SAC;

---

Algorithm 1 overviews the training procedure for LSD. There are two learnable components in LSD: the skill policy $\pi(a|s, z)$ and the representation function $\phi(s)$. In order to impose the 1-Lipschitz constraint on $\phi$, we employ Spectral Normalization (SN) (Miyato et al., 2018). At every epoch, we alternately train $\pi$ with SAC (Haarnoja et al., 2018a) and $\phi$ with stochastic gradient descent (SGD) to maximize Equation (8). When collecting trajectories $\tau$, we fix $z$ within a single episode as in previous works (Eysenbach et al., 2019; Sharma et al., 2020; Kim et al., 2021).

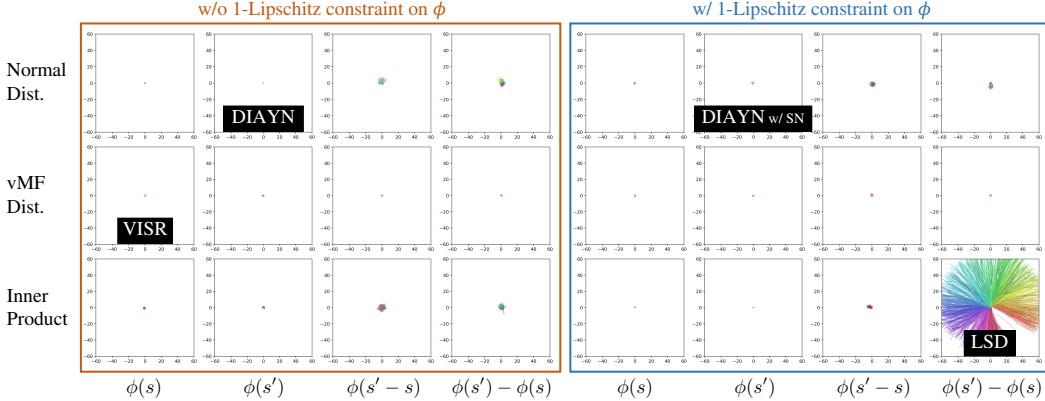

Figure 7: Ablation study on Ant. See Appendix D.

## D    ABLATION STUDY

The LSD reward (Equations (6) and (8)) consists of three components: the inner-product form, the use of $\phi$'s difference and the 1-Lipschitz constraint on $\phi$. In order to examine the role and importance of each component, we ablate them with several variants. We consider all possible combinations of the following variations:

(i) **The form of the reward function** specifying how $z$ and $\phi(\cdot)$ are compared:[1]
   (1) the Normal distribution form (as in DIAYN, Equation (9)), *i.e.*, the squared distance,
   (2) the von-Mises Fisher (vMF) distribution form (as in VISR, Equation (10)), *i.e.*, the inner-product form with normalizing the norms of $\phi(\cdot)$ and $z$, or
   (3) the inner-product form (as in LSD, Equations (8) and (11)) without normalizing the norms.

(ii) **The use of the current and/or next states**: $\phi(s)$, $\phi(s')$ or $\phi(s' - s)$ in place of $\phi(s') - \phi(s)$ (where $s'$ denotes the next state).

(iii) **The use of Spectral Normalization**: with or without the 1-Lipschitz constraint on $\phi(\cdot)$.

Figure 7 shows the result of the ablation study on the Ant environment. We observe the state space coverage drastically decreases if any of these components constituting the LSD reward (Equation (8)) is missing; *i.e.*, all of the three components are crucial for LSD. Especially, just adding Spectral Normalization (SN) to the DIAYN objective ('DIAYN w/ SN', Choi et al. (2021)) does not induce large state variations, since its objective does not necessarily encourage the scale of $\phi(s)$ to increase. We also note that the purposes of using SN in our work and in Choi et al. (2021) are very different. While Choi et al. (2021) employ SN to *regularize* the discriminator for better stability, we use SN to *lower-bound* the state differences (Equation (6)) so that maximizing the LSD objective always guarantees an increase in state differences.

---

[1] For DIAYN (Eysenbach et al., 2019), we omit the $-\log p(z)$ term from the original objective as its expectation can be treated as a constant assuming that $p(z)$ is a fixed prior distribution and that episodes have the same length. For VISR (Hansen et al., 2020), we only consider its unsupervised skill discovery objective.

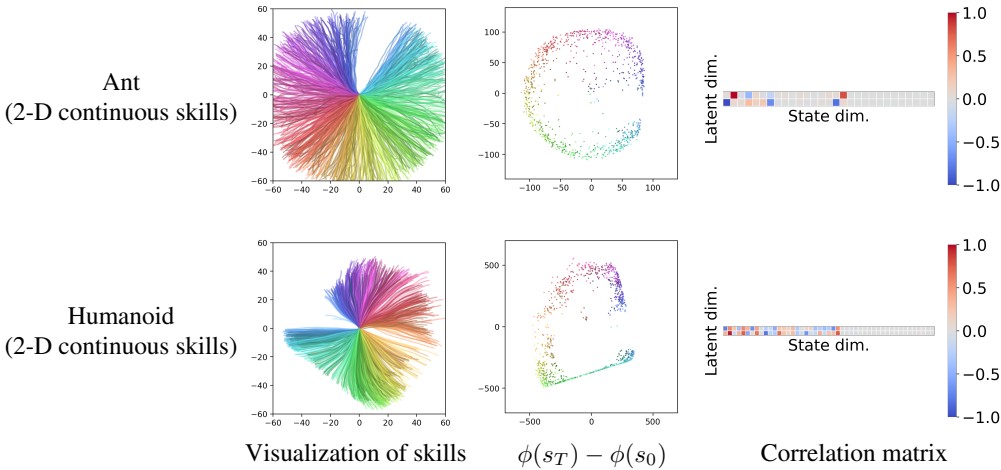

Figure 8: Analysis of 2-D continuous skills discovered by LSD for Ant and Humanoid.

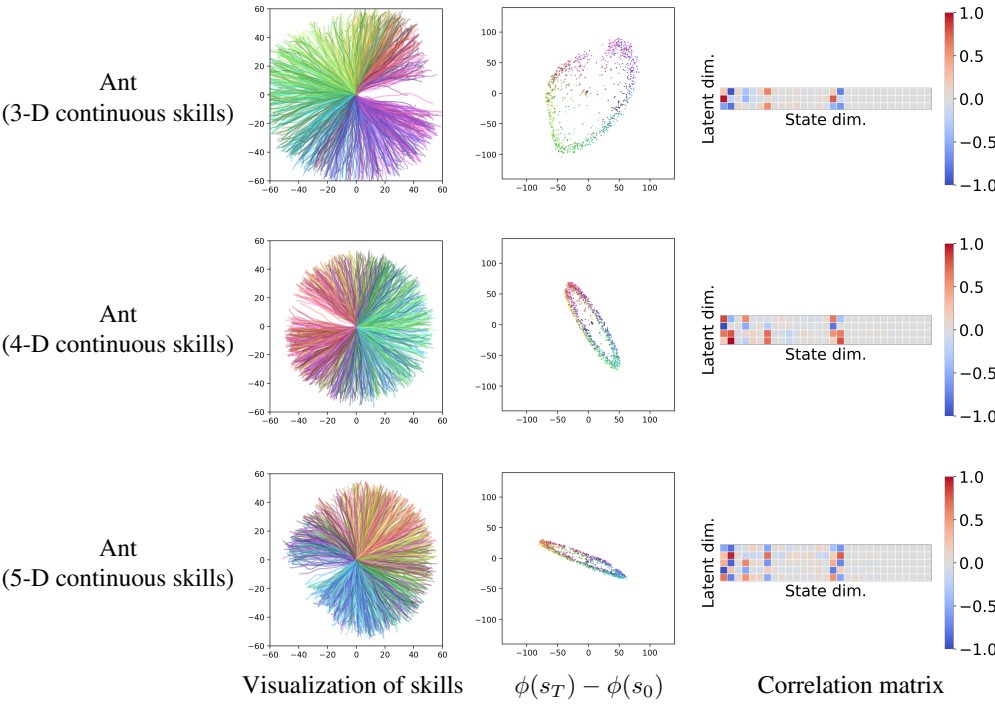

Figure 9: Analysis of 3-D, 4-D and 5-D continuous skills discovered by LSD for Ant. We plot only the first two latent dimensions for the $\phi(s_T) - \phi(s_0)$ figures, and each color also represents only the first two dimensions.

# E   VISUALIZATION AND ANALYSES OF SKILLS LEARNED

We provide more visual examples and analyses of skills discovered by LSD (Figures 8 to 12, also videos available at https://shpark.me/projects/lsd/).

Figure 8 visualizes 2-D continuous skills for Ant and Humanoid, and the learned state representation function $\phi$, and demonstrates the correlation coefficient matrices between the state dimensions and skill latent dimensions. We observe that continuous LSD focuses on the $x$-$y$ coordinates (the first and second state dimensions) on both the environments, which is attributable to the fact that those two

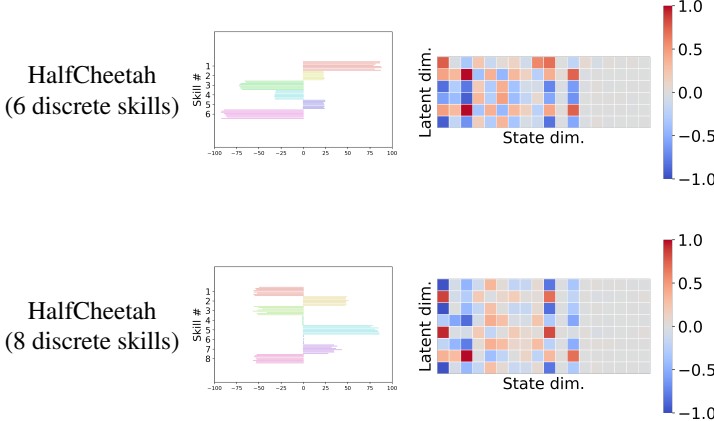

Figure 10: Analysis of discrete skills discovered by LSD for HalfCheetah.

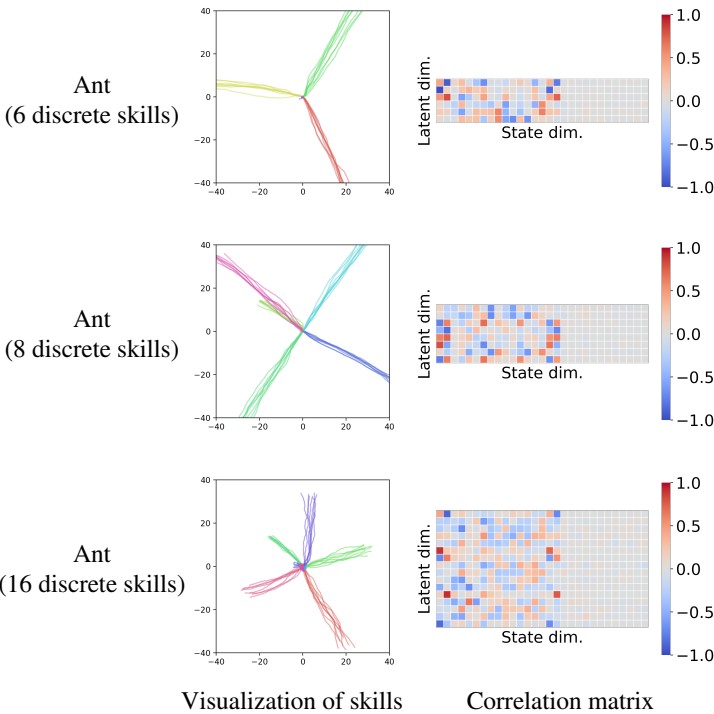

Visualization of skills        Correlation matrix

Figure 11: Analysis of discrete skills discovered by LSD for Ant.

dimensions are the most suitable ones for increasing $\phi(s_T) - \phi(s_0)$ under the Lipschitz constraint. Figure 8 also shows that $\phi$ on Ant has almost no correlation with the 8th to 15th state dimensions, which correspond to the angles of the four leg joints. This is because Ant should repeatedly move its legs back and forth to move consistently in a direction.

We also experiment continuous LSD with $d = 3, 4, 5$ on Ant. Figure 9 demonstrates that LSD still mainly learns locomotion skills as in $d = 2$. However, in this case, some skills represent the same direction since there exist more skill dimensions than $x$ and $y$ dimensions. To resolve this, we believe combining continuous LSD with contrastive learning (as in discrete LSD) can be a possible solution, which we leave for future work.

Figures 10 to 12 visualize discrete skills learned for HalfCheetah, Ant and Humanoid. We confirm that discrete LSD focuses not only on the $x$-$y$ coordinates but on a set of more diverse dimensions.

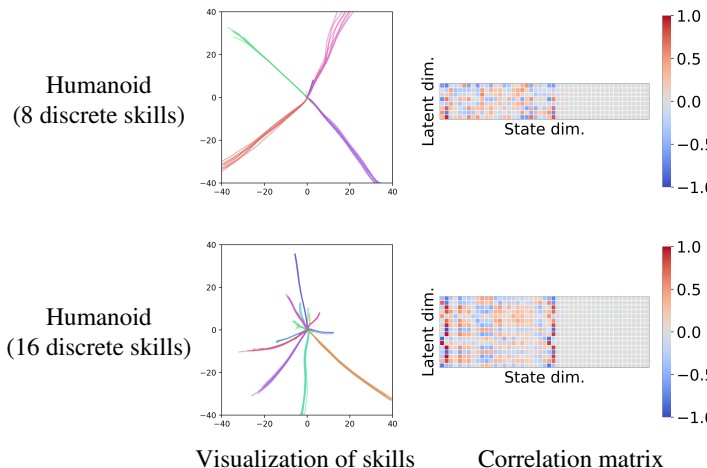

Figure 12: Analysis of discrete skills discovered by LSD for Humanoid.

For instance, the example of 'Ant (16 discrete skills)' in Figure 11 shows that some skills such as the second and third ones have large correlations with the orientation dimensions (the 4th to 7th state dimensions) of Ant. These skills correspond to rotation skills, as shown in the video on our project page.

## F    ENLARGED VISUALIZATION OF SKILL DISCOVERY METHODS

Figure 13 (a zoomed-in version of Figure 2) visualizes the learned skills on the $x$-$y$ plane. Note that each plot has different axes.

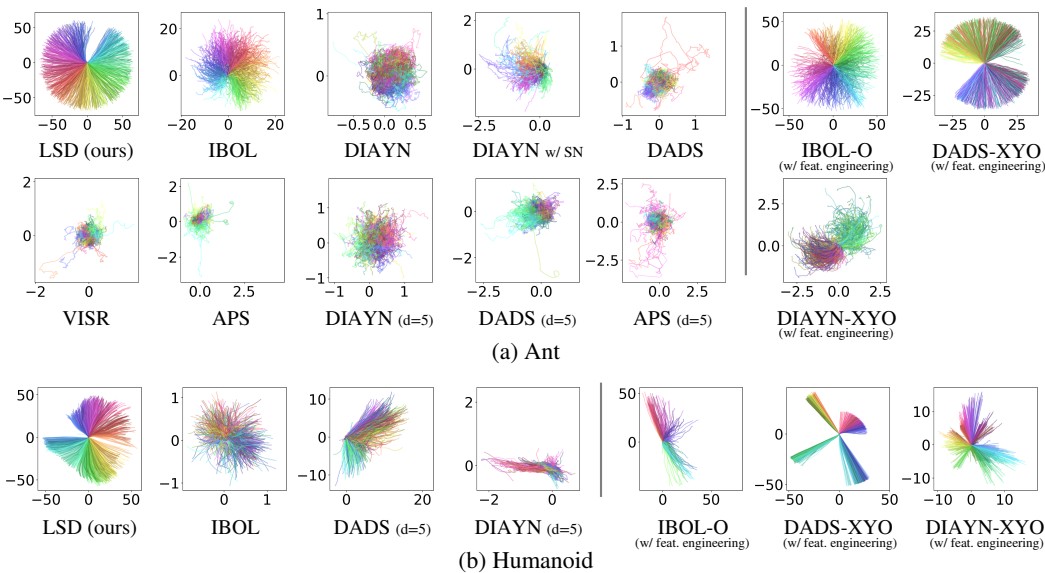

Figure 13: Zoomed-in version of Figure 2. We visualize 2-D (or 5-D) continuous skills discovered by various methods by plotting the $x$-$y$ trajectories of the agent. Each color represents the direction of the skill latent variable $z$. Note that each plot has different axes. This result shows that most of the existing approaches (DIAYN, VISR, APS, DIAYN-XYO, etc.) cannot learn far-reaching locomotion skills, as shown in Figure 17.

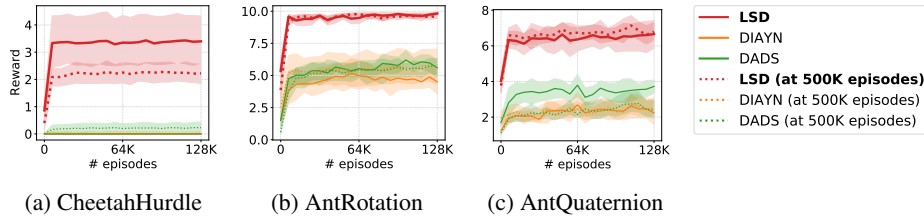

| (a) CheetahHurdle | (b) AntRotation | (c) AntQuaternion |

Figure 14: Training curves of LSD, DIAYN and DADS on three non-locomotion downstream tasks after skill discovery.

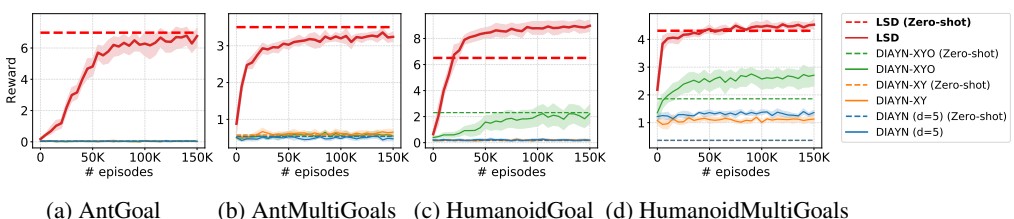

| (a) AntGoal | (b) AntMultiGoals | (c) HumanoidGoal | (d) HumanoidMultiGoals |

Figure 15: Training curves of LSD and DIAYN on four goal-following downstream tasks after skill discovery.

## G  ADDITIONAL DOWNSTREAM TASKS

### G.1  NON-LOCOMOTION DOWNSTREAM TASKS

In order to examine the performance of LSD on more diverse downstream tasks, we make quantitative evaluations of discrete skill discovery methods on non-locomotion environments. Specifically, we test discrete DIAYN, DADS and LSD on three different tasks: CheetahHurdle (with $N = 8$), AntRotation and AntQuaternion (with $N = 16$).

In CheetahHurdle, the HalfCheetah agent should move forward while jumping over evenly spaced hurdles, where we employ the same hurdle configuration used in Qureshi et al. (2020). The agent is given a reward of 1 every time it successfully jumps over a hurdle. In AntRotation, the Ant agent should rotate in place to reach a randomly sampled angle on the $x$-$y$ plane. AntQuaternion is the 3-D version of AntRotation, where the agent should rotate or flip to match a randomly sampled rotation quaternion. In both environments, the agent receives a reward of 10 when the angle between the target orientation and the current orientation becomes smaller than a threshold. Specifically, in AntRotation, we first project both the target angle and the $z$-axis rotation angle of the agent onto the unit circle and compute the Euclidean distance between them. If the distance becomes less than 0.05, the agent gets a reward and the episode ends. In AntQuaternion, we compute the distance between two quaternion using $d(q_1, q_2) = 1 - \langle q_1, q_2 \rangle^2$, where $\langle q_1, q_2 \rangle$ denotes the inner product between the quaternions. When $d(q_1, q_2)$ becomes smaller than 0.3, the agent receives a reward and the episode ends.

**Results.** Figure 14 shows the results on the three downstream tasks, where we report both the performances of skill discovery methods trained with 500K episodes (= 25K epochs) and 2M episodes (= 100K epochs). Figure 14 demonstrates that LSD exhibits the best performance on all of the environments, suggesting that LSD is capable of learning more diverse behaviors other than locomotion skills as well. Also, we observe that the performances of DIAYN and DADS decrease as training of skill discovery progresses in CheetahHurdle. This is mainly because the MI objective ($I(Z; S)$) they use usually prefers more predictable and thus static skills, making the agent incline to posing skills rather than moving or jumping. On the other hand, since LSD's objective always encourages larger state differences, its performance increases with training epochs.

### G.2  ADDITIONAL ZERO-SHOT EVALUATION

In this section, we additionally test the zero-shot scheme for DIAYN-like methods (Choi et al., 2021) on goal-following downstream tasks. DIAYN-like methods can also select a skill to reach a

Table 2: Comparison of zero-shot performances on PointGoal.

| Method | PointGoal ($g_s = 10$) | PointGoal ($g_s = 20$) | PointGoal ($g_s = 40$) | PointGoal ($g_s = 80$) |
|---|---|---|---|---|
| **LSD** | $\mathbf{1.00} \pm 0.00$ | $\mathbf{1.00} \pm 0.00$ | $\mathbf{1.00} \pm 0.00$ | $\mathbf{0.92} \pm 0.03$ |
| DIAYN | $0.41 \pm 0.05$ | $0.20 \pm 0.03$ | $0.12 \pm 0.03$ | $0.05 \pm 0.02$ |
| DIAYN-XYO | $\mathbf{1.00} \pm 0.00$ | $0.80 \pm 0.01$ | $0.35 \pm 0.01$ | $0.15 \pm 0.01$ |

Table 3: Comparison of zero-shot performances on PointMultiGoals.

| Method | PointMultiGoals ($g_m = 10$) | PointMultiGoals ($g_m = 20$) | PointMultiGoals ($g_m = 40$) |
|---|---|---|---|
| **LSD** | $\mathbf{4.00} \pm 0.00$ | $\mathbf{4.00} \pm 0.00$ | $\mathbf{3.85} \pm 0.05$ |
| DIAYN | $1.54 \pm 0.18$ | $0.82 \pm 0.11$ | $0.43 \pm 0.08$ |
| DIAYN-XYO | $2.09 \pm 0.03$ | $1.27 \pm 0.02$ | $0.57 \pm 0.02$ |

target goal $z$ in a zero-shot manner by setting $z = \mathbb{E}_{z'}[q(z'|g)]$, where $q$ denotes its skill discriminator. However, in contrast to LSD, which can follow multiple goals from an arbitrary state with its direction-aligned representation function (Section 3.3), DIAYN-like methods could usually only reach a single goal from the initial state because they are trained to reach the absolute position of the goal with the discriminator $q(\cdot|g)$. Although it may be possible for such methods to deal with unseen goals or initial states if we train them with a broad range of initial states, this modification could harm the performance as it requires more training data. On the other hand, LSD's directional goal scheme enables reaching unseen goals in a zero-shot fashion even if it is trained with a fixed initial state. Also, while they require the skill $z = \mathbb{E}_{z'}[q(z'|g)]$ to be in the vicinity of the prior $p(z)$, LSD is free from this constraint as it can normalize the skill obtained from Equation (12).

### G.2.1 RESULTS ON MUJOCO LOCOMOTION ENVIRONMENTS

Figure 15 demonstrates the performances of LSD, DIAYN and its zero-shot schemes on AntGoal, AntMultiGoals, HumanoidGoal and HumanoidMultiGoals. It suggests that the performance of DIAYN's zero-shot scheme falls behind DIAYN in the HumanoidMultiGoals environment, while LSD's zero-shot performance is mostly comparable to or better than LSD, outperforming DIAYN in all the environments.

### G.2.2 RESULTS ON POINTENV

In order to fairly compare only the zero-shot scheme of each method, we make additional comparisons on a toy environment named PointEnv (Kim et al., 2021). PointEnv is a minimalistic environment in which the state of the agent is defined as its $x$-$y$ position and an action denotes the direction in which the agent moves. If the agent performs action $a = (a_x, a_y) \in [-1, 1]^2$ on state $(s_x, s_y) \in \mathbb{R}^2$, its next state becomes $(s_x + a_x, s_y + a_y)$. Unless otherwise mentioned, the initial state is given as $(0, 0)$. We train each skill discovery method with an episode length of 10 in this environment. We sample $z$ from the 2-D standard normal distribution.

We prepare two goal-following downstream tasks: PointGoal and PointMultiGoals, which are similar to the '-Goal' or '-MultiGoals' environments in Section 4.3. In PointGoal, the agent should reach a goal $g$ uniformly sampled from $[-g_s, g_s]^2$ within 100 environment steps. In PointMultiGoals, the agent should follow four goals within 400 environment steps (we refer to Appendix I.1.5 for the full details of '-MultiGoals' environments). The agent receives a reward of 1 when it reaches a goal.

In these environments, we test the zero-shot schemes of LSD and DIAYN trained with random initial states sampled from $[-10, 10]^2$, as well as DIAYN-XYO with a fixed initial state. When training each skill discovery method, we employ a two-layered MLP with 128 units for modelling trainable components, and train the models for 250K episodes ($= 5K$ epochs) with four minibatches consisting of 500 transitions from 50 trajectories at every epoch. For downstream tasks, we set $g_s \in \{10, 20, 40, 80\}$ and $g_m \in \{10, 20, 40\}$.

Tables 2 and 3 demonstrate the final average zero-shot performance of each skill discovery method, averaged over eight independent runs with the standard error. The result shows that although both LSD and DIAYN-XYO have the same 'base' performance, achieving the maximum reward with $g_s = 10$ (*i.e.*, when goals are sampled only from the states encountered during training), DIAYN-XYO's performance degrades as $g_s$ increases (*i.e.*, given previously unseen goals). We speculate one

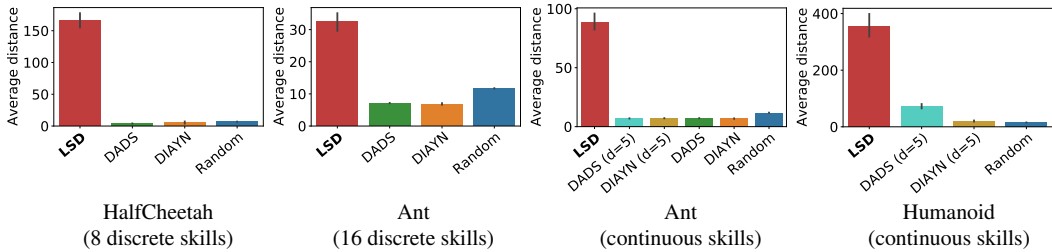

Figure 16: Average normalized state difference $\|s_T - s_0\|$ of skill discovery methods.

reason behind this is that the probability of some chosen skill $z \sim p(z)$ in DIAYN becomes smaller when it encounters a previously unseen goal, which could lead to an unexpected behavior (for example, we notice that the average norm of DIAYN's selected skills is approximately $3.84$ when $g_s = 40$, which is unlikely to be sampled from the standard normal distribution). Also, the result suggests that training with a broad range of the initial state distribution indeed harms the performance of DIAYN. Finally, LSD's zero-shot scheme outperforms DIAYN's on the three PointMultiGoals settings by large margins, indicating that DIAYN's zero-shot scheme (at least empirically) could not cope well with goal-*following* settings (*i.e.*, reaching multiple goals in order).

## H  QUANTITATIVE EVALUATIONS OF $\|s_T - s_0\|$

In order to empirically demonstrate that previous methods using the MI objective $I(Z; S)$ might end up learning only static or simple skills, we measure the average difference between the (normalized) initial and final states $\|s_T - s_0\|$ on MuJoCo environments. Figure 16 shows that existing MI-based methods indeed prefer less dynamic skills, in contrast to LSD. Notably, in HalfCheetah and Ant, the state differences of DIAYN and DADS after skill discovery become even smaller than a random policy. This is natural because it is better for MI-based methods to have more predictable, static trajectories so that they can accurately restore the skill $z$ from the corresponding states (Equation (3)).

## I  IMPLEMENTATION DETAILS

### I.1  MUJOCO LOCOMOTION ENVIRONMENTS

#### I.1.1  SETTINGS

We implement LSD and most of the baselines on top of the garage framework (garage contributors, 2019), while we train the E stage (Lee et al., 2019) of EDL (Campos Camúñez et al., 2020) using their released codebase[2]. We provide the implementation for LSD in the repository at https://vision.snu.ac.kr/projects/lsd/.

For the environments used in our experiments, we use the same configurations adopted in Sharma et al. (2020); Kim et al. (2021), with a maximum episode length of 200 for Ant and HalfCheetah, and 1000 for Humanoid. We normalize each state dimension of the environments ahead of skill discovery with the statistics used in Kim et al. (2021), where they compute the mean and standard deviation from 10000 rollouts of a random policy.

#### I.1.2  VARIANTS OF SKILL DISCOVERY METHODS

DIAYN-XYO (Eysenbach et al., 2019) limits the input to the skill discriminator $q(z|s)$ to the $x$-$y$ dimensions and omits them from the input to its skill policy. For DADS-XYO (Sharma et al., 2020), we make its skill dynamics model $q(s'|s, z)$ only consider the $x$-$y$ coordinates and exclude them from the inputs to both the skill policy and the skill dynamics model, as done in Sharma et al. (2020). IBOL-O (Kim et al., 2021) denotes the exact setting used in their work, where they omit the $x$-$y$ coordinates from the input to its low-level policy.

---

[2]https://github.com/victorcampos7/edl

### I.1.3 TRAINING OF SKILL DISCOVERY METHODS

We model each trainable component as a two-layered MLP with $1024$ units, and train them with SAC (Haarnoja et al., 2018a). At every epoch, we sample 20 (Ant and HalfCheetah) or 5 (Humanoid) rollouts and train the networks with 4 gradient steps computed from 2048-sized mini-batches. For quantitative evaluation of learned skills, we train the models for 2M episodes (= 100K epochs, Ant and HalfCheetah) or 1M episodes (= 200K epochs, Humanoid).

For each method, we search the discount factor $\gamma$ from $\{0.99, 0.995\}$ and the SAC entropy coefficient $\alpha$ from $\{0.003, 0.01, 0.03, 0.1, 0.3, 1.0,$ auto-adjust (Haarnoja et al., 2018b)$\}$. For continuous skills on Ant, we use $\gamma = 0.995$ for the low-level policy of IBOL-O and $0.99$ for the others, and use an auto-adjusted $\alpha$ for DADS, DIAYN, VISR and APS, $\alpha = 0.01$ for LSD, $\alpha = 0.03$ for DADS ($d = 5$), and $\alpha = 0.3$ for DIAYN ($d = 5$). For discrete skills, we set $\alpha$ to $0.003$ (Ant with $N = 16$) or $0.01$ (HalfCheetah with $N = 8$) for LSD and use an auto-adjusted $\alpha$ for DADS and DIAYN. We set the default learning rate to $1e - 4$, but $3e - 5$ for DADS's $q$, DIAYN's $q$ and LSD's $\phi$, and $3e - 4$ for IBOL's low-level policy. On Ant and HalfCheetah, we train the models with on-policy samples without using the replay buffer (following Sharma et al. (2020); Kim et al. (2021)), while we use the replay buffer for sampling the $k(= 5)$-nearest neighbors in APS. For the low-level policy of IBOL-O, we additionally normalize rewards and use full-sized batches, following Kim et al. (2021). In the case of IBOL, we used mini-batches of size 1024 and do not normalize rewards as we find this setting performs better. We use the original hyperparameter choices for their high-level policies.

On Humanoid, we set the discount factor to $\gamma = 0.99$ and the learning rate to $3e - 4$, but $1e - 4$ for DADS's $q$, DIAYN's $q$ and LSD's $\phi$. Also, we use the replay buffer and additionally give an alive bonus $b$ at each step (following Kim et al. (2021)) searched from $\{0, 0.03, 0.3\}$, while we find that $b$ does not significantly affect the performance. We use $\alpha = 0.03$ for LSD and an auto-adjusted $\alpha$ for the others, and $b = 0$ for DADS and $b = 0.03$ for the others.

### I.1.4 TRAINING OF SMM

We train the SMM exploration stage (Lee et al., 2019) of EDL (Campos Camúñez et al., 2020) using the official implementation with hyperparameters tuned. Specifically, we set the discount factor $\gamma$ to $0.99$ (Ant and Humanoid), the SAC entropy coefficient $\alpha$ to $1$ (Ant) or $3$ (Humanoid), the $\beta$-VAE coefficient to $1$, the alive bonus $b$ to $30$ (only for Humanoid), and the density coefficient $\nu$ to $0.5$ (Ant) or $0.05$ (Humanoid).

### I.1.5 DOWNSTREAM TASKS

In AntGoal and HumanoidGoal, a random goal $g \in [-g_s, g_s]^2$ is given at the beginning of each episode, and the episode ends if the agent reaches the goal (*i.e.*, the distance between the agent and the goal becomes less than or equal to $\epsilon$). In AntMultiGoal and HumanoidMultiGoals, (up to $X$) new goals are sampled within the relative range of $[-g_m, g_m]^2$ from the current 2-D coordinates when the episode begins, the agent reaches the current goal, or the agent does not reach the goal within $Y$ steps. In all of the environments, the agent receives a reward of $r$ when it reaches the goal (no reward otherwise). We use $g_s = 50$, $g_m = 15$, $\epsilon = 3$, $X = 4$ and $Y = 50$ for AntGoal and AntMultiGoals, and $g_s = 20$, $g_m = 7.5$, $\epsilon = 3$, $X = 4$ and $Y = 250$ for HumanoidGoal and HumanoidMultiGoals. Also, we set $r$ to $10$ ('-Goals' environments) or $2.5$ ('-MultiGoals' environments).

### I.1.6 TRAINING OF HIERARCHICAL POLICIES FOR DOWNSTREAM TASKS

For downstream tasks, we train a high-level meta-controller on top of a pre-trained skill policy with SAC (Haarnoja et al., 2018a) for continuous skills or PPO (Schulman et al., 2017) for discrete skills. The meta-controller is modeled as an MLP with two hidden layers of 512 dimensions. We set $K$ to 25 (Ant and HalfCheetah) or 125 (Humanoid), the learning rate to $3e - 4$, the discount factor to $0.995$, and use an auto-adjusted entropy coefficient (SAC) or an entropy coefficient of $0.01$ (PPO). For SAC, we sample ten trajectories and train the networks with four SAC gradients computed from full-sized batches at every epoch. For PPO, we use 64 trajectories, ten gradients and 256-sized mini-batches. We normalize the state dimensions with an exponential moving average. For zero-shot learning, we set $g$ in Equation (12) to the current state with its locomotion dimensions replaced with the goal's coordinates. Additionally, only for Figure 1c, we set the locomotion dimensions of the input to the pre-trained low-level policy to 0 for better visualization.

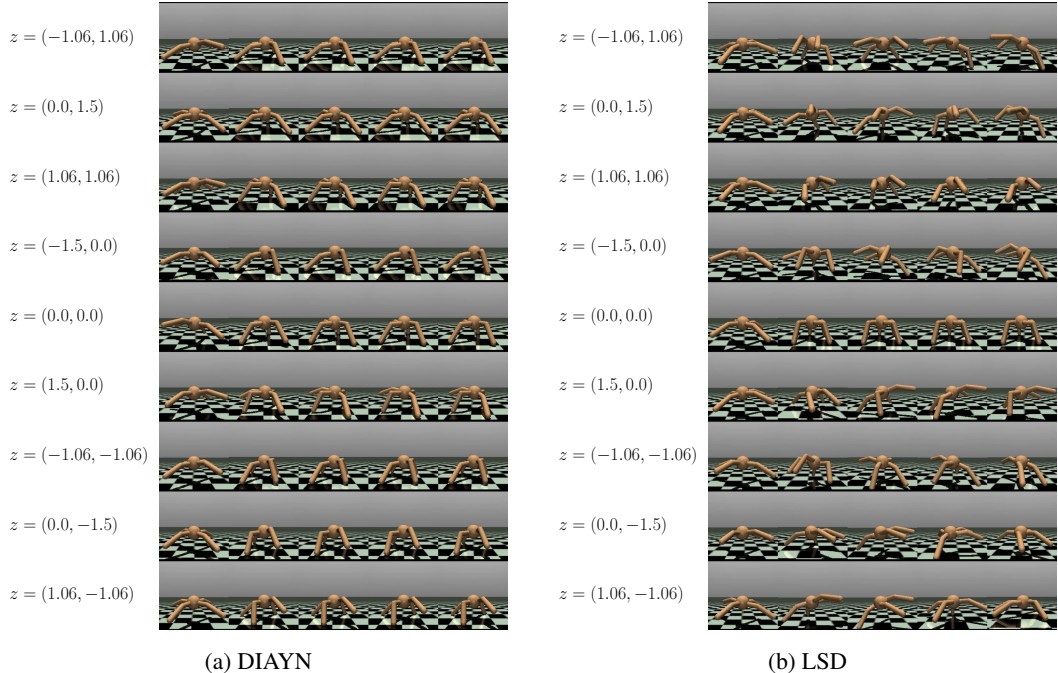

$z = (-1.06, 1.06)$     $z = (-1.06, 1.06)$

$z = (0.0, 1.5)$     $z = (0.0, 1.5)$

$z = (1.06, 1.06)$     $z = (1.06, 1.06)$

$z = (-1.5, 0.0)$     $z = (-1.5, 0.0)$

$z = (0.0, 0.0)$     $z = (0.0, 0.0)$

$z = (1.5, 0.0)$     $z = (1.5, 0.0)$

$z = (-1.06, -1.06)$     $z = (-1.06, -1.06)$

$z = (0.0, -1.5)$     $z = (0.0, -1.5)$

$z = (1.06, -1.06)$     $z = (1.06, -1.06)$

(a) DIAYN          (b) LSD

Figure 17: 2-D continuous skills for Ant. DIAYN discovers posing skills, as its mutual information objective does not necessarily prefer large state variations. On the other hand, LSD encourages the agent to have more variations in the state space, resulting in learning more dynamic behaviors such as locomotion skills. Videos are available on our project page.

## I.2 MuJoCo Manipulation Environments

### I.2.1 Training of Skill Discovery Methods

For MuJoCo manipulation environments (FetchPush, FetchSlide, FetchPickAndPlace), we implement skill discovery methods based on the official implementation[3] of MUSIC (Zhao et al., 2021). We train each method for 8K episodes (= 4K epochs) with SAC and set the model dimensionality to $(1024, 1024)$, the entropy coefficient to $0.02$, the discount factor to $0.95$ and the learning rate to $0.001$. At every epoch, we sample two trajectories and train the models with $40$ gradient steps computed from $256$-sized mini-batches. For methods equipped with the MUSIC intrinsic reward, we set the MUSIC reward coefficient to $5000$ with reward clipping, following Zhao et al. (2021). For the skill reward coefficient, we perform hyperparameter search over $\{5, 15, 50, 150, 500, 1500, 5000\}$, where we choose $500$ (LSD), $150$ (DADS), or $1500$ (DIAYN), not clipping the skill reward.

### I.2.2 Training of Downstream Policies

In FetchPushGoal, FetchSlideGoal and FetchPickAndPlaceGoal, a random goal is sampled at the beginning of each episode and the episode ends with a reward of $1$ if the agent reaches the goal, where we use the same sampling range and reach radius as the original Fetch environments. For training of meta-controllers, we use the same hyperparameters as in the skill discovery phase, except that we sample $16$ trajectories for each epoch.

## J Rendered Scenes of Learned Skills

Figures 17 to 21 visualize the rendered scenes of skills discovered on the MuJoCo locomotion environments. Each figure demonstrates a *single* set of skills. We refer the reader to our project page for the videos.

---

[3] https://github.com/ruizhaogit/music

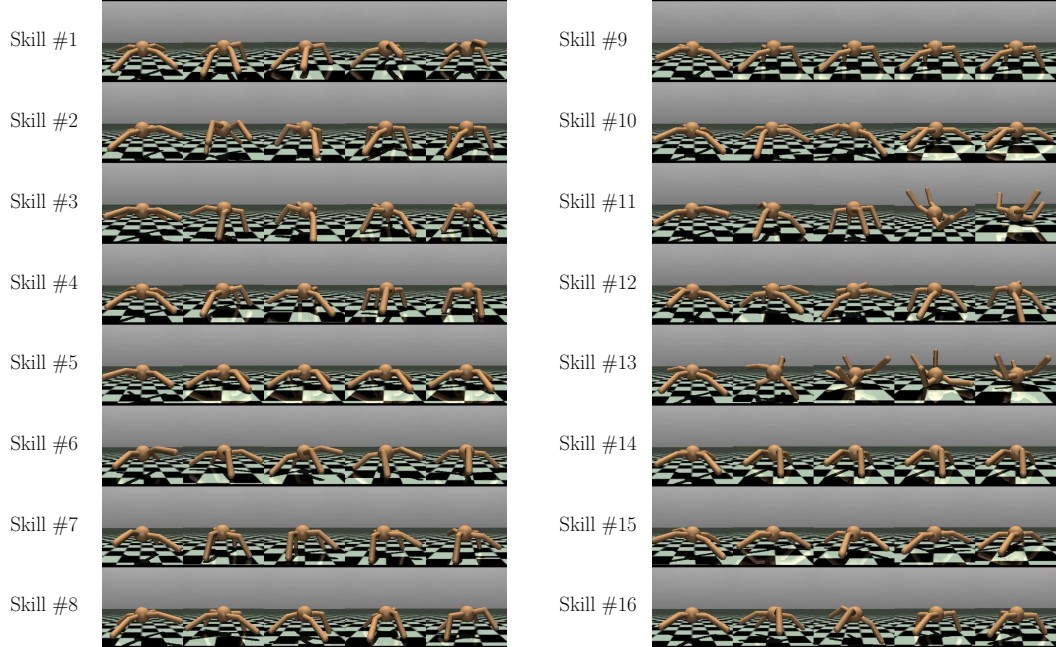

Figure 18: 16 discrete skills discovered by LSD for Ant. Discrete LSD learns a skill set consisting of locomotion skills (#1, #6, #7, #12, #16), rotation skills (#2, #3, #4, #8, #10, #15), posing skills (#5, #9, #14) and flipping skills (#11, #13). Videos are available on our project page.

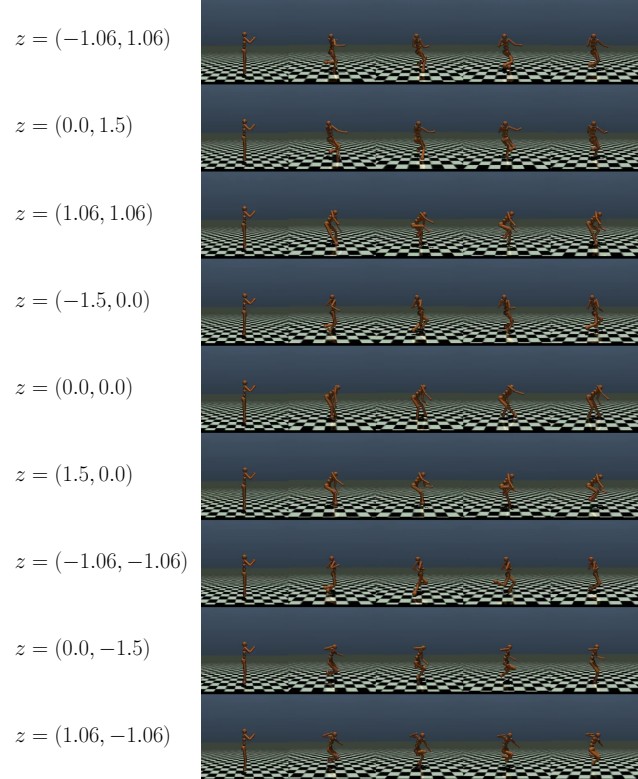

Figure 19: 2-D continuous skills discovered by LSD for Humanoid. The Humanoid robot can walk in various directions specified by $z$. Videos are available on our project page.

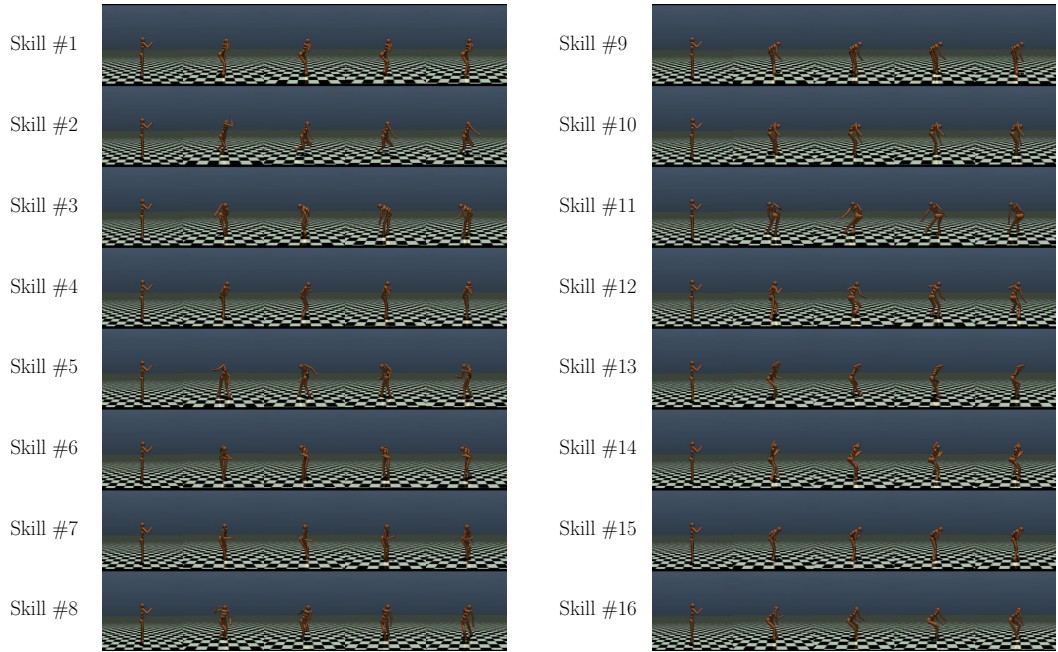

Figure 20: 16 discrete skills discovered by LSD for Humanoid. Videos are available on our project page.

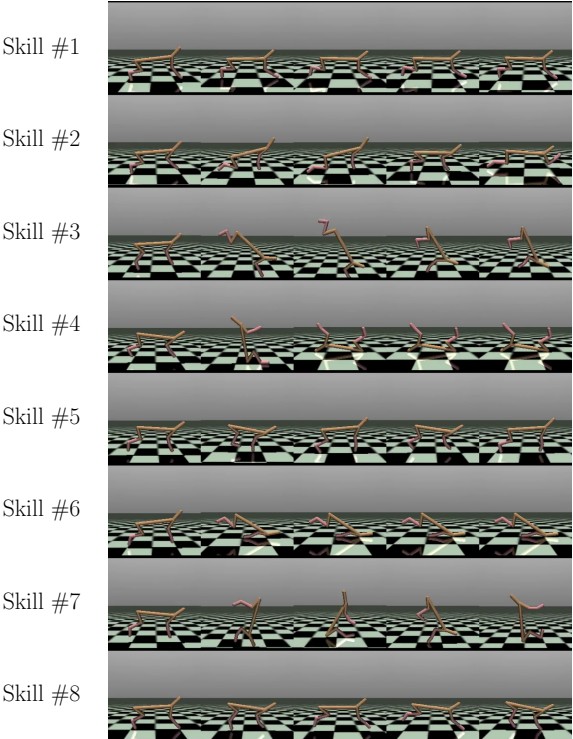

Figure 21: 8 discrete skills discovered by LSD for HalfCheetah. Videos are available on our project page.

