# OpenReview forum: "Lipschitz-constrained Unsupervised Skill Discovery"
_ICLR.cc/2022/Conference — ICLR 2022 Poster_

### Official Review · Reviewer_8wK8 · 2021-10-21

**Correctness:** 4
**Technical Novelty And Significance:** 3
**Empirical Novelty And Significance:** 4
**Recommendation:** 6
**Confidence:** 4

**Main Review:**


The paper proposes a sensible method with empirically very significant results. Further, interesting theoretical connections to existing work are provided and the paper is generally well written with high-quality evaluation.

However, tzhe paper also suffers from several issues
1. The proposed method is ad-hoc. See Sec 3.2, “Specifically, we suggest replacing the implicit regularizer of VIC (the second term in Equation (5)) with a 1-Lipschitz constraint on φ”, which is done without justification. The connection to mutual information skills is intriguing, but the proposed method is not shown to maximize mutual information.
2. The “discrete” version of the method is even more ad-hoc. Indeed, it is impossible since a discrete variable cannot be Lipschitz-continuous. Instead, the continuous method is used while restricting the prior skill distribution to a finite number of skills. It is unclear what are the theoretical properties of this method since it is different from the standard methods that use discrete skill variables.
3. The introduction is very confusing and does not state the actual contribution of the paper. Instead, the claimed contributions 1 and 2 in the intro are not novel. Both constraining the skill encoder to be Lipschitz continuous and using skill discovery for zero-shot evaluation was proposed by Choi’21 (or arguably by even earlier work). Why does the contribution paragraph not focus on differences with prior work?
4. The paper claims that none of the prior work can perform zero-shot evaluation, while also citing Choi'21 that explains the opposite. Please provide zero-shot evaluation for the prior work that the paper compares to.


---- Update -----

The updated paper resolves most of my initial concerns, therefore I increase my score to accept. However, I am not confident that this method will scale to visual environments as the method crucially relies on Eucledian distance in the observation space to be meaningful (via the Lipschitz constraint), as the authors themselves note in the limitations section. Further, it appears from Fig 9 that the method ignores some of the state dimensions entirely and therefore might not be able to learn diverse skills that involve all state dimensions such as reaching a desired joint pose. Therefore I am not able to give a strong accept.



**Summary Of The Paper:**


The paper proposes a new skill discovery objective based on lipschitz-constrained dynamics skills. Specifically, for a state encoding \phi, <\phi_{t+1} - \phi_t, z> is maximized with a DIAYN-style training, and spectral normalization on \phi. The connection to DIAYN with Gaussian skills is shown, which is an unconstrained version of the proposed objective. A version of the method that defines a prior distribution of skills with finite support is also presented, which can be used similarly to discrete skill learning methods. Empirically, the method achieves significant improvements over prior work, being able to tackle ant and humanoid domains, where all prior methods fail.


**Summary Of The Review:**

The paper proposes a novel method with significant performance improvement. However, the paper suffers from several issues with accuracy of the claims. Further, a version of the method dubbed "discrete Lipschitz-constrained skill discovery" is proposed, which is clearly a contradiction (discrete variables cannot be Lipschitz constrained). The paper needs to provide a clear explanation of what the discrete version of the method actually is. Without fixing these issues, specifically points 2-4 above, I believe the paper cannot be accepted.

---

> ### Author Response · Authors · 2021-11-20
> **Author Response to Reviewer 8wK8 (1/2)**
>
> We appreciate the reviewer’s thoughtful and constructive feedback.
>
> **1. The proposed method is not shown to maximize mutual information**
>
> We improved the manuscript (Section 3.1) to further clarify our motivation behind continuous LSD. The main point is that our goal is *not* to maximize $I(Z; S)$ (or its lower bound), but to propose a new, different objective apart from the existing MI-based objectives. As a result, we can overcome the limitation of the MI objective that can be maximized even with very small state variation.
>
> We also note that replacing the second term in Equation (5) with a 1-Lipschitz constraint is neither to approximate nor to modify the MI objective. Our intent is to use only the *directional* component (the first term) to overcome the drawback of the MI objective. However, without the Lipschitz constraint, it could be trivially maximized by just infinitely increasing the value of $\phi(s)$, without encouraging any state variation. Thus, adding a Lipschitz constraint is a natural (not an ad-hoc) choice, which leads to our new objective of Equation (6). We revised Section 3.2 in the draft to clarify this point.
>
>
> **2. Discrete variables cannot be Lipschitz constrained, which is clearly a contradiction**
>
> We would like to respectfully correct the misunderstanding. Please note that the 1-Lipschitz constraint is imposed *only* on the state representation function $\phi(s)$, i.e., $\| \phi(s’) - \phi(s) \| \leq \| s’ - s \|$, which takes only *continuous* state $s$. It is not imposed on the discrete skill variable $z$. Therefore, there was no contradiction in our original submission. We have made this clear in the manuscript (Section 3.4).
>
>
>
> **3. Discrete LSD is ad-hoc, and is different from the standard methods that use discrete skill variables**
>
> As the reviewer mentioned, discrete LSD is different from previous discrete skill discovery methods in that it uses a modified encoding instead of the standard one-hot encoding for discrete skills. The rationale behind this choice is not ad-hoc but rather logical, but we apologize that this was not clearly described earlier.
>
> As in revised Section 3.4, for discrete skills, we cannot directly use the standard one-hot encoding in our LSD objective due to the fact that it is not *zero-centered*. If we use a non-zero-centered encoding for discrete LSD in Equation (6), it can have a failure mode where all the skills may be collapsed into a single behavior (please see below for a more detailed description).
>
> One natural way to prevent such collapse is to make the standard one-hot encoding have a zero mean (i.e., to normalize it), which directly leads to our modified scheme in Equation (13). Then, it also provides the following intuitive interpretation.
>
> $r\_k^{\text{LSD}}(s\_t, a\_t, s\_{t+1}) = [\phi(s\_{t+1}) - \phi(s\_t)]\_k - \frac{1}{N-1}\sum\_{i \in \{1,2,\ldots,N\} \setminus \{k\}} [\phi(s\_{t+1}) - \phi(s\_t)]\_i$
>
>
> In this equation (Equation (14)), the first term increases the $k$-th dimension of $\phi(s\_{t+1}) - \phi(s\_t)$ while the second term decreases the other dimensions of it, which can push the learned skills away from each other: i.e., it learns discrete skills in a *contrastive* manner.
>
> *Why can’t we use a non-zero-centered encoding like the standard one-hot vectors?*
>
> (Partially quoted from the updated draft) Without loss of generality, suppose that the first dimension of the mean of the $N$ encoding vectors is $c > 0$. Then, if the agent finds a final state $s_T$ that makes $\| s_T - s_0 \|$ fairly large, it can simply learn the skill policy to reach $s_T$ regardless of skills $z$ so that the agent can always receive a reward of $c \cdot \|s_T - s_0\|$ on average, by setting, e.g., $\phi(s_T) = [\|s_T - s_0\|, 0, \ldots, 0]^\top$ and $\phi(s_0) = [0, 0, \ldots, 0]$. In other words, the agent can easily exploit the reward function without learning any diverse set of skills. However, this will not happen when the skill encoding is zero-centered (i.e., $c=0$). Finally, the reason why existing methods do not suffer from this failure mode is that they use a categorical distribution (unlike LSD), where its softmax function implicitly has a similar role to our contrastive scheme.
>
> We hope this response with revised Section 3.4 provides a better insight into our design choice on the encoding of discrete skill variables.

---

> > ### Author Response · Authors · 2021-11-20
> > **Author Response to Reviewer 8wK8 (2/2)**
> >
> > **4. Novelty in the contribution 1: Comparison with Choi et al. (2021) in terms of spectral normalization**
> >
> > Our first contribution is that we propose a *new* skill discovery objective (different from the existing MI objective) based on a Lipschitz constraint, which mainly focuses on the directions in the latent space. This encourages the agent to discover skills that have large variations in the state space, unlike previous MI-based methods.
> >
> > Although Choi et al. (2021) also make use of spectral normalization (SN), they still rely on the MI objective $I(Z;S)$, applying SN as a regularizer. Thus, they still suffer from the limitation of the MI objective, which does *not* necessarily encourage large state variations (Appendix H). On the other hand, we propose a different objective that departs from the MI objective. Our ablation study (Figure 7 in Appendix D) demonstrates that just imposing a 1-Lipschitz constraint on the discriminator (Choi et al. 2021) does not result in learning far-reaching locomotion behaviors.
> >
> > Moreover, our purpose of using a Lipschitz constraint is also very different from Choi et al. (2021). They employ SN to *regularize* the discriminator for better stability (therefore their MI objective can still work without the constraint). However, we use the Lipschitz constraint to *lower-bound* the state differences so that maximizing the LSD objective guarantees an increase in state differences as well. Please see our response to Q2 of Reviewer wpgt for a more detailed explanation.
> >
> > We have made this point clearer in the updated draft (Introduction, Section 3.2 and Appendix D).
> >
> >
> > **5. Novelty in the contribution 2: Regarding zero-shot evaluation or goal-reaching**
> >
> > The reviewer is correct that previous DIAYN-like approaches (i.e., methods that use the identity $I(Z;S) = H(Z) - H(Z|S)$) can also select a single skill $z = \mathbb{E}[q(\cdot|g)]$ to reach a goal state $g$ from the initial state $s_0$, where $q$ denotes the skill discriminator.
> >
> > However, it is usually *not* possible for DIAYN-like approaches to reach a given goal from an arbitrary state, or to follow multiple goals, in a zero-shot manner. Instead, it can only make a single transition that has been encountered during training. This is because DIAYN-like methods are trained to reach the *absolute* position of the goal specified by $z$ from the *initial* state. Their $z = \mathbb{E}[q(\cdot|g)]$ depends only on the target goal $g$, with no consideration of its current position. On the other hand, LSD learns to move in the *direction* in the latent space designated by $z$; thus, it can follow multiple goals in order (as in Figure 1c) or reach a goal from an *arbitrary* state, exploiting our learned representation function $\phi$ (Equation (12)).
> >
> > To clarify this point, we revised the draft as follows. We emphasized the difference between the two schemes (Section 3.3). Also, following the reviewer’s suggestion, we added an additional experiment on DIAYN and its zero-shot scheme in Appendix G.2. The result in Figure 15 suggests that LSD and its zero-shot variant still outperform other baselines by a large margin.
> >
> >
> > We appreciate the reviewer’s time and feedback to improve our paper, and hope that our response with the revised draft would address the concerns. We would be happy to answer any further questions.

---

> > > ### Comment · Reviewer_8wK8 · 2021-11-22
> > > **Response by 8wK8**
> > >
> > > Thanks for the detailed reply. However, I still feel like some of the issues are not addressed. Specifically, the main remaining issue is that the claims in the contribution are still too strong and the newly added experiment in the appendix appears to contradict them.
> > >
> > > > 1,2,3.
> > >
> > > Perhaps the main reason the proposed method seems ad-hoc is because the LSD objective appears to be derived as an ad-hoc modification to MI in section 3.2. If the intent was to instead introduce a different framework based on "state encoders" rather than "discriminators", perhaps it would be better to present the new method first and the connection to MI-based objective second. That would clarify what 'Lipschitz-constrained' means and that the method is not derived from MI. It should also be clarified that the latent space is simply an encoding of the state rather than a skill space.
> > >
> > > > 4. Novelty in the contribution 1: Comparison with Choi et al. (2021) in terms of spectral normalization
> > >
> > > I agree that the method is different from Choi'21. However, the intro unfortunately still does not explain how. Specifically, it is stated that the method is designed to "discover skills with large state variations", but the same could be said about Choi'21 because it is not clear what "state variations" means. I think this could be fixed with a minor rewording e.g. by talking about "traveled distances in the state space" instead, similar to the eigenoptions literature.
> > >
> > > > 5
> > >
> > > Thanks for additional experiment, however, I am still confused about the claim in the intro that DIAYN cannot reach goals zero-shot, even though the additional experiments seems to show that DIAYN does that somewhat well.
> > >
> > > In addition, it is not clear to me what is the claimed deficiency of DIAYN in the multi-goal context. Since DIAYN gets the skill simply from the skill encoder, it should be easy to obtain the skill for the second goal. Is the claim that LSD somehow generalizes better to new states which allows it to reach goals from different initial states? If so, I don't think there is a strong theoretical or empirical argument for such generalization in the paper currently. Moreover, this implies that DIAYN could be improved simply by using a wide initial state distribution for training. So, the claim that DIAYN does not solve tasks zero-shot needs to be weakened.

---

> > > > ### Author Response · Authors · 2021-11-23
> > > > **Author Response to Reviewer 8wK8 (1/3)**
> > > >
> > > > We genuinely appreciate the reviewer’s additional feedback and active engagement in the discussion. We hope that the response below and the updated manuscript fully address the additional concerns by the reviewer.
> > > >
> > > >
> > > > **1, 2, 3. Clarifying the derivation of the LSD objective**
> > > >
> > > > Thanks for the great suggestion. We agree that the motivation and derivation of the LSD objective can become clearer by decoupling it from its connection with the MI objectives. Following the advice, we have updated the manuscript to introduce the decomposition of the MI objective (Section 3.1) and the derivation of the LSD objective (Section 3.2) separately.
> > > >
> > > > **4. Rewording "state variations"**
> > > >
> > > > We agree with the reviewer that the suggested expression, "traveled distances", explains the difference of LSD from prior works such as Choi et al. (2021) more precisely. Thus, we have revamped our manuscript to minimize possible confusion and clarify the difference, accordingly.
> > > >
> > > > **5. Regarding LSD’s zero-shot goal-following**
> > > >
> > > > First, we would like to elaborate on the difference between LSD and DIAYN-like methods in terms of zero-shot goal-following.
> > > >
> > > > (a) While DIAYN's zero-shot scheme (Choi et al., 2021) relies on that the skill $z = \mathbb{E}[q(\cdot|g)]$ (for some goal $g$ encountered during training) lies in the vicinity of the prior $p(z)$ (as we previously mentioned in Appendix G.2), LSD does not since it only cares about directions in the latent space, which could enable reaching a goal that is not encountered during training data. For example, in a goal-reaching environment, if DIAYN tries to reach a goal at $(x=40, y=0)$ in a zero-shot manner, it will select a skill as $\mu(g=(40, 0))$ (where we assume that $\mu(g)$ is the mean of the $q(\cdot|g)$ distribution). However, if the goal $(40, 0)$ is not encountered during its training phase, $\mu(g=(40, 0))$ could output a highly unlikely skill according to the prior $p(z)$ and thus to the policy, which may make the agent perform an unexpected behavior (please see below for empirical demonstrations). On the other hand, since LSD only cares about *directions* in the latent space, we can freely normalize a selected skill (Equation (12)) to move in the desired direction to a given goal, which enables further generalization to unseen goals.
> > > >
> > > > (b) Although the standard DIAYN could reach a previously encountered goal from the fixed initial state used for training in a zero-shot manner (which also accounts for its non-zero performance in HumanoidMultiGoals in Figure 15), it is usually not able to reach a given goal from an arbitrary state or to follow multiple goals. For example, let us assume that DIAYN learns directional locomotion skills with the fixed initial state $(0, 0)$ and we want to follow two goals $g = (2, 0)$ and $(1, 0)$ in order. Then, even if DIAYN successfully reaches the first goal $g = (2, 0)$, when it tries to reach the second goal, it could still move right as $z = \mu(g=(1, 0))$ would hardly be a skill of moving left (since it is trained with the fixed initial state $(0, 0)$, and the goal $(1, 0)$ is placed on the right side of $(0, 0)$). Nevertheless, as the reviewer pointed out, DIAYN may be able to learn to reach a goal from an arbitrary state if it is trained with a wide initial state distribution. However, we argue that since such a modification would require more training data, it could hinder the training and result in even a lower performance compared to the current one, which already significantly falls behind LSD (Figure 3, 15) on MuJoCo locomotion environments (please see below for clearer demonstrations of this on toy environments). In contrast, thanks to our directional scheme, it is empirically shown that LSD could select an appropriate skill to reach a goal from an arbitrary state with Equation (12) in a zero-shot manner, even if it is trained with a fixed initial state. The similar performances of LSD and LSD (zero-shot) on AntMultiGoals and HumanoidMultiGoals (Figure 15) substantiate this.

---

> > > > > ### Author Response · Authors · 2021-11-23
> > > > > **Author Response to Reviewer 8wK8 (2/3)**
> > > > >
> > > > > In order to clearly demonstrate the differences between the two schemes, we perform additional experiments on a toy environment named *PointEnv*. The reason why we test on the toy environment is that the (already existing) substantial performance differences between LSD and DIAYN (or DIAYN-XYO) on the MuJoCo Ant and Humanoid environments make it very difficult to dissect and fairly compare only their zero-shot schemes.
> > > > >
> > > > > (Partially quoted from the updated manuscript) PointEnv is a minimalistic environment where the state of the agent (a point) is defined as its x-y position, and an action denotes the vector in which the agent moves. If the agent performs action $a = (a_x, a_y) \in [-1, 1]^2$ on state $(s_x, s_y) \in \mathbb{R}^2$, its next state becomes $(s_x + a_x, s_y + a_y)$. Unless otherwise mentioned, the initial state is given as $(0, 0)$. We train each skill discovery method with an episode length of 10 in this environment. We sample $z$ from the 2-D standard normal distribution.
> > > > >
> > > > > We also prepare two goal-following downstream tasks: PointGoal and PointMultiGoals, which are similar to the ‘-Goal’ or ‘-MultiGoals’ environments in the manuscript. In PointGoal, the agent should reach a goal $g$ uniformly sampled from $[-g_s, g_s]^2$ within 100 environment steps. In PointMultiGoals, the agent should follow four goals sampled from $[-g_m, g_m]^2$ (based on the current coordinates) within 400 environment steps (we refer to Appendix I.1.5 for the full details of ‘-MultiGoals’ environments). The agent receives a reward of 1 when it reaches a goal.
> > > > >
> > > > > In these environments, following your suggestion, we test the zero-shot schemes of LSD and DIAYN trained with random initial states sampled from $[-10, 10]^2$. We also include DIAYN-XYO in our experiments. For downstream tasks, we test with $g_s \in \\{10, 20, 40, 80\\}$ and $g_m \in \\{10, 20, 40\\}$.
> > > > >
> > > > >
> > > > > | Method | PointGoal ($g\_s = 10$) | PointGoal ($g\_s = 20$) | PointGoal ($g\_s = 40$) | PointGoal ($g\_s = 80$) | PointMultiGoals ($g\_m = 10$) | PointMultiGoals ($g\_m = 20$) | PointMultiGoals ($g\_m = 40$) |
> > > > > |:-|-:|-:|-:|-:|-:|-:|-:|
> > > > > | LSD | $\textbf{1.00} \pm 0.00$ | $\textbf{1.00} \pm 0.00$ | $\textbf{1.00} \pm 0.00$ | $\textbf{0.92} \pm 0.03$ | $\textbf{4.00} \pm 0.00$ | $\textbf{4.00} \pm 0.00$ | $\textbf{3.85} \pm 0.05$ |
> > > > > | DIAYN | $0.41 \pm 0.05$ | $0.20 \pm 0.03$ | $0.12 \pm 0.03$ | $0.05 \pm 0.02$ | $1.54 \pm 0.18$ | $0.82 \pm 0.11$ | $0.43 \pm 0.08$ |
> > > > > | DIAYN-XYO | $\textbf{1.00} \pm 0.00$ | $0.80 \pm 0.01$ | $0.35 \pm 0.01$ | $0.15 \pm 0.01$ | $2.09 \pm 0.03$ | $1.27 \pm 0.02$ | $0.57 \pm 0.02$ |
> > > > >
> > > > > The table above demonstrates the final average zero-shot performance of 8 independent runs of each skill discovery method with the standard error.
> > > > > It shows that although both LSD and DIAYN-XYO have the same ‘base’ performance, achieving the maximum reward with $g_s = 10$ (i.e., when goals are sampled only from the states encountered during training), DIAYN-XYO’s performance degrades as $g_s$ increases (i.e., given previously unseen goals). We speculate one reason behind this is that the probability of some chosen skill $p(z)$ in DIAYN becomes smaller when it encounters a previously unseen goal, which could lead to unexpected behavior (for example, we notice that the average norm of DIAYN’s zero-shot skills is approximately 3.84 when $g_s = 40$, which is unlikely to be sampled from the standard normal distribution). Also, the result suggests that training with a broad range of the initial state distribution indeed harms the performance of DIAYN. Finally, LSD’s zero-shot scheme outperforms DIAYN’s on the three PointMultiGoals settings by large margins, indicating that DIAYN’s zero-shot scheme (at least empirically) could not cope well with goal-*following* settings (i.e., reaching multiple goals in order).

---

> > > > > > ### Author Response · Authors · 2021-11-23
> > > > > > **Author Response to Reviewer 8wK8 (3/3)**
> > > > > >
> > > > > > To summarize, our claimed benefit of LSD over DIAYN-like methods is that it can efﬁciently solve goal-following tasks with a wider range of goals (i.e., following multiple goals (even unseen during training) from an arbitrary state) in a zero-shot fashion compared to previous methods, thanks to our direction-aligned state representation $\phi$. We have empirically demonstrated this via (i) that the performance of LSD (zero-shot) mostly matches LSD on multiple-goal-following locomotion tasks, outperforming any other baselines by a large margin (Figure 5 (AntMultiGoals and HumanoidMultiGoals)) and (ii) that LSD (zero-shot) performs better than DIAYN (zero-shot) (Choi et al., 2021) on PointEnv in reaching previously unseen goals.
> > > > > >
> > > > > > Nonetheless, we acknowledge that our zero-shot scheme and its generalizability would not be perfect nor applicable to all possible downstream goal-following environments (although it empirically demonstrates strong performance in our experiments). Also, we agree with the reviewer that the claim that DIAYN-like methods cannot solve goal-reaching tasks in a zero-shot is somewhat overstated. Regarding this point, we have thoroughly revised the manuscript (please see the updated contributions, Table 1, Section 3.3 and Appendix G.2).

---

> > > > > > > ### Comment · Reviewer_8wK8 · 2021-11-23
> > > > > > > **Response by 8wK8**
> > > > > > >
> > > > > > > Thank you for the extensive revisions and new experiments! I believe they make the paper significantly stronger. I have updated my score.

---

> ### Author Response · Authors · 2021-11-26
> **Thanks for the update!**
>
> Thank you so much for helping improve our work with active participation in the discussion and reconsidering the score!
>
> Just as a side note, we would like to highlight that discrete LSD can learn a more diverse set of behaviors than continuous LSD, such as skills reaching specific joint poses, moving, or flipping in Ant (Figure 18), taking into consideration most of the non-velocity dimensions, e.g., positions, joint angles, and orientations (Figure 10-12) (the other state dimensions, which discrete LSD ignores, are the velocity features that depend on the former dimensions).
>
> We again deeply appreciate your effort and time put into the review.

---

### Official Review · Reviewer_QUmk · 2021-11-02

**Correctness:** 4
**Technical Novelty And Significance:** 3
**Empirical Novelty And Significance:** 4
**Recommendation:** 6
**Confidence:** 3

**Main Review:**

Strengths:


The paper is well written and motivated. The discussion in section 3.2 is intuitive and makes the decisions behind the LSD objective quite clear.


The experimental results are quite impressive, in Ant and Humanoid particularly, and it seems that LSD is quite effective at discovering more dynamic skills compared to prior work, without specific domain knowledge.


The downstream task performance of LSD (using a hierarchical controller which selects actions in the space of skills) is a significant improvement compared to a number of sensible prior works.


The method is relatively simple to implement (as an additional reward to a model-free RL method), and an implementation is provided for reproducibility.

Weaknesses:

The related work by Choi et al also uses spectral normalization to improve learned skill quality, so it needs to be better clarified how this work stands with respect to that.

As the authors mention in the conclusion, LSD assumes that the Lipschitz constraint is meaningful ($ell_2$ norms in the state space are meaningful) which is not always true, for example when learning from pixels, or perhaps in some more complex manipulation settings. But I agree that representation learning would likely help to address this issue when learning from pixels.

The scale of figure 2 seems like it could be slightly misleading: while it makes sense that LSD would generate skills which cover the state space well (especially in the state magnitude), it’s unclear if the other discovered skills are also able to perform the same action (maybe the ant can still walk, but just not as far). For example, DIAYN-XYO on the Ant environment may learn some locomotion skills but its trajectories are almost invisible on the figure, which makes it seem like it doesn’t learn any skills.


**Summary Of The Paper:**

The paper introduces Lipschitz-constraint skill discovery (LSD), a method for unsupervised skill discovery. The method addresses one limitation of the line of unsupervised skill discovery works which use mutual information between states produced by a skill and latent skill representations as the training objective, which is that they often discover overly simple and static skills, and require some additional domain information to be provided to learn more dynamic skills. LSD addresses this problem by reformulating a Lipschitz-constrained objective to learn continuous or discrete skills, and then demonstrates the effectiveness of the learned skills for downstream tasks.


**Summary Of The Review:**

I think the paper can be accepted with some modifications. The method is well-motivated, tackling an important issue with mutual-information based skill discovery methods. It demonstrates impressive qualitative and quantitative results compared to prior works. The paper reads very well and is easy to follow.

---

> ### Author Response · Authors · 2021-11-20
> **Author Response to Reviewer QUmk**
>
> We appreciate the reviewer’s thoughtful and constructive feedback.
>
> **1. How does LSD compare with Choi et al. (2021) with respect to the use of Spectral Normalization**
>
> As the reviewer points out, both our work and Choi et al. (2021) make use of spectral normalization (SN). Choi et al. use SN for better stability and smoothness by limiting the expressivity of the discriminator. However, our use of SN (or a 1-Lipschitz constraint) is fundamentally different as follows. First, we use the 1-Lipschitz constraint to ensure that the maximization of our objective always entails an increase in state variations (Equation (6)). Consequently, unlike Choi et al. where SN is used only as a regularizer and thus it can be dropped, LSD’s objective does not operate without the 1-Lipschitz constraint (Appendix D). Moreover, the key motivation of our objective (Equation (6)) is the divorce from the MI objective $I(Z; S)$ that Choi et al. still try to optimize. Following the suggestion, we have described this difference more clearly in the revised manuscript (Section 3.2 and Appendix D).
>
> For an empirical demonstration of the difference, we provide an ablation study in Appendix D. We tested all 24 combinations of (i) whether to use SN or not, (ii) three different reward function forms and (iii) four different current/next state combinations. Figure 7 in Appendix D shows that only LSD can successfully discover locomotion skills, suggesting that just adding Spectral Normalization on DIAYN as in Choi et al. (2021) is not sufficient. This is because the MI lower-bound objective (even with Spectral Normalization) does not necessarily encourage large variations, in contrast to LSD.
>
> **2. Clarification on Figure 2**
>
> The trajectories in Figure 2 that look like a dot (DIAYN, DIAYN-XYO, etc.) indicate that the learned behaviors are of limited or no movements. In particular, DIAYN-XYO equipped with the x-y prior still learns no meaningful locomotion skills. It instead learns skills of slightly moving or rotating, resulting in very small displacement. Again, this is because the existing MI objective (or discriminability) can still be maximized with such small differences in states. For clarification, we added a zoomed-in version of Figure 2 in Appendix F (Figure 13).
>
>
> We appreciate the reviewer’s time and feedback to improve our paper, and hope that our response with the revised draft would address the concerns. We would be happy to answer any further questions.

---

### Official Review · Reviewer_wpgt · 2021-11-02

**Correctness:** 4
**Technical Novelty And Significance:** 4
**Empirical Novelty And Significance:** 4
**Recommendation:** 8
**Confidence:** 3

**Main Review:**

Figure 5 seems to show that LSD gets fundamentally better skills as competing methods look like they are asympoting to a lower level in at least AntMultiGoal, HumanoidGoal and HumanoidMultiGoal. The evaluation on classic AI Gym as well as robot pick and place tasks shows good scalability to a variety of interesting tasks.

The objective looks simpler to evaluate than a number of the competing approaches promising a more efficient algorithm.

The idea of aligning the direction of latent and observed states is nice for downstream planning but will this ultimately limit the flexibility of the latent states to express important properties of the space that may not be one-to-one?

In equation 6, is the <= sign backwards? Shouldn’t the projected states be larger || phi(x)-phi(y) || >= ||x-y|| according to the argument of the paper?

“LSD” and “learning” including “machine learning” has a big literature already that might make it hard to find the “LSD” paper. Maybe something like Lipschitz-constrained Unsupervised Skill Acquisition (LUSA)?

While LSD technically doesn’t make use of hand engineered features, the fact that the algorithm enforces a directionality consistent with the observed states pace kind of builds in a similar concept to maximizing travel in space … it is a bit subtler but not completely different. The case here seems overstated.

**Summary Of The Paper:**

The paper builds upon the DIAYN idea (Eysenback 2018) that an agent could develop skills in an unsupervised environment by finding a set of skills that collectively visits the whole state space but encourages each skill to cover a different subspace and later use one of these skills to simplify the learning of a downstream task. The paper points out that prior work maximizes the distribution of latent states, but does not necessarily maximize world state differences, because latent states are the result of a learned projection from world states. The paper introduces an explicit Lipschitz constraint that forces latent state differences to be larger than observed state differences and relaxes a prior constraint of equality so that the latent states only need to move in the same direction as observed states but not in a fixed ratio. The paper shows this can be implemented as an efficient per step intrinsic reward for unsupervised learning.  Since the latent state aligns with observed state, one can use simple planner on downstream tasks to choose the right z to achieve a goal in observed space without any additional learning or fine tuning (zero shot skill exploitation). The paper evaluates LSD on 2D skills in the widely used AI GYM benchmark. Qualitatively, the LSD model discovers challenging dynamic behaviors in the difficult humanoid environment that competing unsupervised models fail to discover. Quantitatively, expected reward is higher for LSD and LSD zero shot than competing models. Direct evaluation of state space coverage shows LSD has significantly larger coverage than competing models supporting the paper hypothesis. LSD is also demonstrated on robotic tasks and is able to learn to pick objects in using unsupervised learning.

**Summary Of The Review:**

The paper proposed a new practically implementable objective for unsupervised learning that gives qualitatively and quantitatively better performance on classic AI Gym benchmarks and challenging robotic pick and place.

---

> ### Author Response · Authors · 2021-11-20
> **Author Response to Reviewer wpgt**
>
> We appreciate the reviewer’s thoughtful and constructive feedback.
>
> **1. Limited flexibility of latent states**
>
> We agree that aligning only the directions (not considering the magnitudes) of latent states in continuous LSD might limit the flexibility of latent mapping. However, it can be advantageous in downstream applications since it enables reaching multiple goals in a zero-shot manner. Thanks to this alignment, we can select skills by simply considering their directions, with no need to predict distances. Also, given that learned skills are often later combined to solve different downstream tasks, a high-level controller that uses those skills can be trained to control how long the selected skill should last in the environment. This can compensate for the reduced flexibility. Nonetheless, we believe making continuous LSD consider both the directions and magnitudes can be another interesting future direction.
>
> **2. Regarding Equation (6)**
>
> We believe the $\leq$ sign in Equation (6) is correct in its current form:
>                       $\| \phi(s’) - \phi(s) \| \leq \| s’ - s \|$,
> which is precisely the definition of a 1-Lipschitz constraint. Intuitively, our goal is to ensure that maximizing the latent differences “pushes” the states away from one another. This is achieved as follows: When we maximize the objective of Equation (6), $J^{\text{LSD}} = \mathbb{E}_{z,\tau} \left[ (\phi(s_T) - \phi(s_0))^\top z \right]$, the length of $\phi(s_T) - \phi(s_0)$ would increase. Since $\| \phi(x) - \phi(y) \| \leq \|x - y\|$ always holds due to the 1-Lipschitz constraint, it then “pushes” $s_T$ away from $s_0$, which eventually leads to learning skills that encourage large differences of state. In short, we maximize the *lower bound* of $s_T - s_0$.
>
> **3. Regarding the name of the method ‘LSD’**
>
> Thanks for the suggestion! To avoid possible confusion, we won’t be updating the method name during the rebuttal period, but will consider renaming (such as LUSA, LCSD, etc.) in the camera-ready version.
>
> **4. LSD not using hand-engineered features seems overstated**
>
> We agree with the reviewer that while LSD does not use any explicit feature engineering, maximizing the latent differences (and thus state differences) entails a subtle inductive bias for the resulting skills. Although we normalize the state dimensions so that LSD can operate independently of the initial scales of state dimensions, LSD’s skills still depend on maximally reachable regions in the normalized state space, as the reviewer pointed out. We have toned down some strong statements in the main paper, and clarified this in the limitations section.
>
>
> We appreciate the reviewer’s time and feedback to improve our paper, and hope that our response with the revised draft would address the concerns. We would be happy to answer any further questions.

---

### Official Review · Reviewer_an2L · 2021-11-05

**Correctness:** 4
**Technical Novelty And Significance:** 3
**Empirical Novelty And Significance:** 3
**Recommendation:** 6
**Confidence:** 3

**Main Review:**

Strengths:
- Clear write-up with good motivation that correctly points out a common failure mode of previous MI-based objectives.
- Good performance on basic and challenging robots, including Humanoid. Overall, the empirical evaluation is quite exhaustive, and multiple relevant baselines are included
- Alignment of representation and skill enables zero-shot goal reaching, which is nicely demonstrated.

Weaknesses:
- A lot of focus is put on skills resulting in directed locomotion. While this is a useful result and allows to easily measure coverage, for example, it's also a limited evaluation -- for directed locomotion, it's pretty straightforward to simply learn goal-directed policies and use them as skills. It would be nice to consider other tasks as well, e.g. letting the Cheetah jump over hurdles (as in DIAYN, for example). It's good that the authors also evaluate on robot manipulation tasks (Fetch* environments), but if I understood correctly, in these setting they consider the location of the target object as the only state features for reward and state representation?
- The authors acknowledge that with continuous skills, their method mostly discovers locomotion skills. I'm left wondering to what degree this is caused by the skill dimension being set to 2. What happens if 3 or more dimensions are used?
- The notion of "dynamic skill", which is used quite a bit in the abstract and motivation, is not really clear. It's finally explained in section 3.1., but I don't find it very intuitive.

**Summary Of The Paper:**

This paper proposes a new objective (LSD) for unsupervised skill discovery, building on prior skill discovery work using mutual information criteria. As in previous works, skills are parameterized by a latent variable. The main ingredients are changes to the MI-based reward that (1) maximize alignment of learned state representation and latent variable, and (2) ensuring that state space differences are larger in the actual state space than in the learned representation. These modifications act as a prior so that the skill variable contains semantics (as it encodes a direction in representation space) and skills are encouraged to significantly change the agent's state.

**Summary Of The Review:**

Overall I think this is a good paper, but also that the focus on the navigation tasks is rather narrow and that it would be nice to a few non-navigation downstream tasks with the standard MuJoCo agents.

---

> ### Author Response · Authors · 2021-11-20
> **Author Response to Reviewer an2L**
>
> We appreciate the reviewer’s thoughtful and constructive feedback.
>
> **1. A lot of focus is put on skills resulting in directed locomotion**
>
> As the reviewer pointed out, we made quantitative comparisons (e.g., state space coverage, downstream tasks) mainly on locomotion tasks. This is partly because previous skill discovery methods (continuous DIAYN, DADS, APS, IBOL and LSD) mostly learn locomotion or posing skills on the MuJoCo control benchmark.
>
> Following the reviewer’s suggestion, we have made additional comparisons of discrete skill discovery methods on tasks going beyond directed locomotion. Specifically, we compare discrete DIAYN, DADS and LSD on three non-locomotion tasks: **CheetahHurdle** (with $N=8$), **AntRotation** and **AntQuaternion** (with $N=16$).
>
> (1) In CheetahHurdle, the HalfCheetah agent should jump over evenly spaced hurdles.
>
> (2) In AntRotation, the Ant agent should rotate to reach a randomly sampled angle on the $x$-$y$ plane.
>
> (3) AntQuaternion is the 3-D version of AntRotation, where the agent should rotate and/or flip to reach a randomly sampled rotation quaternion.
> We refer to Appendix G.1 for further details of the environments.
>
> | Method | CheetahHurdle (500K) | CheetahHurdle (2M)  | AntRotation         | AntQuaternion       |
> | :-     | -:                   | -:                  | -:                  | -:                  |
> | **LSD**    | $\textbf{2.22} \pm 0.19$  | $\textbf{3.40} \pm 0.49$ | $\textbf{9.83} \pm 0.08$ | $\textbf{6.66} \pm 0.43$ |
> | DIAYN  |   $0.03 \pm 0.03$    |   $0.00 \pm 0.00$   |   $4.51 \pm 0.49$   |   $2.24 \pm 0.36$   |
> | DADS   |   $0.23 \pm 0.10$    |   $0.00 \pm 0.00$   |   $5.59 \pm 0.35$   |   $3.73 \pm 0.20$   |
>
> The number following $\pm$ denotes the standard error of the final rewards from 8 independent runs. The results show that LSD outperforms other baselines by large margins, mainly because only LSD encourages larger variations in the state space (not necessarily limited to the $x$-$y$ dimensions), which leads to learning a variety of skills such as jumping or flipping. Also, the performances of DIAYN and DADS on CheetahHurdle decrease with more training iterations, as shown in the performances at 500K and 2M episodes. It could be because their MI maximization objective inclines to more predictable and thus static skills. We provide the full results and further explanation in Appendix G.1.
>
> **2. Continuous LSD with $d \ge 3$**
>
> This is a great question. We additionally test LSD with $d = 3, 4, 5$ on Ant, and report the result in Appendix E. While each skill dimension has different correlation coefficients with respect to the state dimensions, they primarily focus on locomotion dimensions and thus the agent mainly learns to move as diverse directions as in $d = 2$. However, since the number of skill dimensions is larger than 2 ($x$ and $y$), LSD would map multiple skills to the same direction. As we found that discrete LSD exhibits better skill diversity than continuous LSD, we believe combining continuous LSD with a contrastive scheme (as in discrete LSD) could be a possible approach for learning a more high-dimensional manifold, which we leave for future work.
>
> **3. The notion of ‘dynamic’ skills is not really clear until section 3.1**
>
> We have clarified this point in Section 1, explicitly mentioning that ‘dynamic’ means ‘making large state variations’.
>
>
> We appreciate the reviewer’s time and feedback to improve our paper, and hope that our response with the revised draft would address the concerns. We would be happy to answer any further questions.

---

### Decision · Program_Chairs · 2022-01-20

**Decision:**

Accept (Poster)

**Comment:**

The paper introduces unsupervised skill discovery using Lipschitz-constrained skills. It is well-written and demonstrates the advantages in a solid experimental section.